# Asymmetric activation mechanism of a homodimeric red light-regulated photoreceptor

**Geoffrey Gourinchas[1], Udo Heintz[2], Andreas Winkler[1]\***

[1]Institute of Biochemistry, Graz University of Technology, Graz, Austria; [2]Max Planck Institute for Medical Research, Heidelberg, Germany

**Abstract** Organisms adapt to environmental cues using diverse signaling networks. In order to sense and integrate light for regulating various biological functions, photoreceptor proteins have evolved in a modular way. This modularity is targeted in the development of optogenetic tools enabling the control of cellular events with high spatiotemporal precision. However, the limited understanding of signaling mechanisms impedes the rational design of innovative photoreceptor-effector couples. Here, we reveal molecular details of signal transduction in phytochrome-regulated diguanylyl cyclases. Asymmetric structural changes of the full-length homodimer result in a functional heterodimer featuring two different photoactivation states. Structural changes around the cofactors result in a quasi-translational rearrangement of the distant coiled-coil sensor-effector linker. Eventually, this regulates enzymatic activity by modulating the dimer interface of the output domains. Considering the importance of phytochrome heterodimerization in plant signaling, our mechanistic details of asymmetric photoactivation in a bacterial system reveal novel aspects of the evolutionary adaptation of phytochromes.

DOI: https://doi.org/10.7554/eLife.34815.001

**\*For correspondence:**
andreas.winkler@tugraz.at

**Competing interests:** The authors declare that no competing interests exist.

While symmetry is a recurring feature of life on earth, many processes in biology deviate from this general trend. Starting from the selection of specific building blocks for biological systems, that is *L*-amino acids and *D*-sugars for proteins and nucleic acids, respectively, asymmetry has impacted living organisms ever since the origin of life. The importance of asymmetric processes ranges from the regulation of enzymes (*Kim et al., 2017*) and signaling processes (*Sysoeva et al., 2013*; *Neiditch et al., 2006*; *Alvarado et al., 2010*), to the polarity (*Li and Gundersen, 2008*) and division (*Knoblich, 2001*) of cells, and all the way to phenotypic asymmetry of whole animals (*Levin, 2005*). Specifically in the field of sensor-effector communication, asymmetry is emerging as a critical aspect of both signal integration (*Kim et al., 2017*; *Neiditch et al., 2006*; *Narayanan et al., 2014*; *Chervitz and Falke, 1996*) and regulation of various output functionalities (*Kim et al., 2017*; *Moore and Hendrickson, 2012*; *Hu et al., 2013*).

An interesting group of environmental sensors are light-regulated photoreceptors; proteins that interact with various cofactors to convey light responsiveness to different qualities and intensities of light (*Fraikin et al., 2013*). These photoreceptors control complex processes in all kingdoms of life ranging from phototaxis in prokaryotes (*Gomelsky and Hoff, 2011*) to intricate regulatory networks of photomorphogenesis in plants (*Fraser et al., 2016*) or entrainment of circadian rhythms in animals (*Cashmore, 2003*). As far as naturally occurring light regulated functionalities are concerned, current evidence for asymmetric properties of the frequently oligomeric photoreceptor systems is mainly observed in the family of plant phytochromes where two different types of asymmetry have been implicated in providing flexible light-regulated systems. The first one is a consequence of the evolutionary duplication of phytochrome genes and the observation that some family members only form mixed heterodimers, for example *Arabidopsis thaliana* phyC and phyE only form heterodimers with

phyB (*Mathews and McBreen, 2008*; *Clack et al., 2009*; *Liu and Sharrock, 2013*). The second form of asymmetry relates to the functional state of each individual light responsive module in either classical homodimeric or mixed heterodimeric red light photoreceptors. Phytochromes generally interconvert between an inactive red-light absorbing ($P_r$) and a physiologically active far-red-light absorbing ($P_{fr}$) species. Different biophysical properties of functional heterodimers featuring a $P_{fr}/P_r$ assembly have been suggested as one plausible model to explain differences in morphological development at different photon flux densities of actinic light (*Casal et al., 1998*; *Klose et al., 2015*; *Hennig and Schäfer, 2001*), extending previous models that range from differences in degradation (*Clough and Vierstra, 1997*) to interactions among different phytochrome family members (*Sharrock and Clack, 2004*).

Our characterization of the bacterial phytochrome *Is*PadC (phytochrome activated diguanylyl cyclase from *Idiomarina species A28L*) shows that structural rearrangements at the coiled-coil sensor-effector linker that increase enzymatic activity are coupled to heterodimer formation characterized by two different functional phytochrome states. Based on the functional implications of a quasi-translational switching between two coiled-coil conformations coupled to asymmetric changes of the sensory modules, such a mechanism of sensor-effector cross-talk might be more widespread than currently believed, both in phytochrome-like protein signaling and in regulatory mechanisms of dimeric enzymatic functionalities. In canonical phytochromes such asymmetric heterodimers feature one dark-adapted bilin conformation ($P_r$, cofactor in its *ZZZssa* configuration [*Murgida et al., 2007*]) and one activated $P_{fr}$ state with bilin isomerized at the C15-C16 methine bridge (*ZZEssa* [*Salewski et al., 2013*]). Changes in the local environment of the cofactor affect the whole photosensory module (PSM) since the bilin chromophore interacts with different regions of the PSM. In bacterial phytochromes the bilin chromophore is covalently linked via a thioether linkage to a conserved cysteine residue located in the N-terminal extension (NTE), an α-helix preceding the N-terminal Period/ARNT/Single-minded (PAS) domain (*Lamparter et al., 2004*). The main stabilization of the open-chain tetrapyrrole cofactor is mediated by a cGMP phosphodiesterase/adenylyl cyclase/FhlA (GAF) domain following the PAS domain. The complete PSM of canonical phytochromes features an additional phytochrome-specific (PHY) domain at the C-terminus that stabilizes the photoactivated $P_{fr}$ state via residues from a domain extension, the PHY-tongue, that interacts with the bilin binding pocket in the GAF domain and undergoes a conformational rearrangement from β-hairpin to α-helix upon $P_r$-to-$P_{fr}$ photoconversion (*Takala et al., 2014*; *Yang et al., 2009*; *Anders et al., 2013*; *Burgie et al., 2016*).

Here, we show how structural rearrangements in the vicinity of the cofactors of a functional $P_{fr}/P_r$ heterodimer affect additional functional elements of the PSM and how these asymmetric dimers influence downstream signaling partners on a molecular level. Specifically, $P_{fr}$ formation is coupled to a quasi-translational rearrangement of one helix in the coiled-coil sensor-effector linker element that enables switching between two conformational states. While these molecular details are important for the understanding of bacterial diguanylyl cyclase regulation, they also have far reaching implications for the structurally related nucleotidyl cyclases, where the importance of the preceding helical elements has only recently been appreciated (*Ziegler et al., 2017*; *Lindner et al., 2017*; *Etzl et al., 2018*) and linked to the occurrence of genetic diseases (*Vercellino et al., 2017*). In comparison to functional details of other characterized phytochrome systems that form active $P_{fr}/P_{fr}$ homodimers, our results reveal an interesting molecular mechanism for the evolutionary adaptability of various sensor-effector systems. Considering that the rational design of novel sensor-effector combinations is a critical aspect for the rapidly developing field of optogenetics, the consideration of asymmetric functional states will be beneficial for understanding and optimizing initial designs of promising new light-regulated functionalities.

## Results

### Screening and expression of *Is*PadC variants stabilized in different coiled-coil registers

Our previous characterization of the *Is*PadC system revealed that the phytochrome – diguanylyl cyclase (DGC) linker region populates two different coiled-coil register conformations that influence the enzymatic activity of the effector (*Gourinchas et al., 2017*). To test this hypothesis, amino acids

at special positions of the coiled-coil were replaced to either destabilize or stabilize the presumably stimulating register conformation (*Figure 1a*). Previously, the corresponding sequence variants of the coiled-coil register were used in screening assays - *Is*PadC$^{Reg1}$ (*Is*PadC D504L A518L; constitutively inactive) and *Is*PadC$^{Reg2}$ (*Is*PadC S505V A526V; constitutively active) – and a detailed biochemical characterization was only provided for variants with parts of the coiled-coil deleted (*Gourinchas et al., 2017*). This time we focused entirely on the sequence variants with identical linker length and included an additional construct, *Is*PadC$^{Reg2.a}$ (*Is*PadC S505V N512V N519V A526V), that is further stabilized in the stimulating register. To this end, we substituted the highly conserved, polar asparagine residues at the generally hydrophobic *a* positions of the coiled-coil in its stimulating register 2 conformation with valine residues (*Figure 1* and *Figure 1—figure supplement 1a–c*). Employing a Congo red-based screening system for cyclic-dimeric-GMP production in *E.coli* (*Antoniani et al., 2010*), we show that *Is*PadC$^{Reg2.a}$, comparable to *Is*PadC$^{Reg2}$, is constitutively active under both dark and light conditions, whereas *Is*PadC$^{Reg1}$ is always inactive (*Figure 1b*). All three constructs were expressed in an *E. coli* strain coexpressing heme oxygenase to isolate holoproteins loaded with the biliverdin co-factor and to enable a more detailed biochemical characterization. It should be noted that so far none of the *Is*PadC constructs tested could be purified as apoprotein as they appear to irreversibly aggregate inside the cells in the absence of their cofactor. Therefore, all final protein preparations correspond to fully loaded holoproteins of each construct. A three-step purification procedure, consisting of an affinity, a reverse affinity and a size-exclusion chromatography step, allowed isolation of all full-length proteins to apparent homogeneity. Based on the elution volume of the purified proteins in the size exclusion step, all variants form stable dimers in solution. All subsequent biochemical and biophysical experiments were performed with protein samples obtained by this procedure.

## Individual protomers of dimeric *Is*PadC feature different biliverdin environments upon illumination

During our previous characterization of *Is*PadC, we observed only partial formation of the P$_{fr}$ state even under constant red light illumination (*Gourinchas et al., 2017*), which clearly differs from absorption spectra of other well-characterized bacteriophytochromes, such as for example *Deinococcus radiodurans* phytochrome (*Wagner et al., 2005*; *Wagner et al., 2007*) (*Dr*BphP, *Figure 1c*), or bathyphytochromes that have P$_{fr}$ as their ground state (*Yang et al., 2009*; *2011*). In an attempt to better understand the molecular basis for the incomplete P$_{fr}$ spectrum under steady-state red light conditions, we analyzed the effect of actinic light intensity and the protein concentration on the light-state spectrum. As shown in *Figure 1d* decreasing protein concentrations correlate with a reduction of the required light intensity to increase the 750 nm contribution, however, in all cases similar steady-state light-activated spectra of *Is*PadC are populated. Importantly, the sigmoidal profile obtained for activation of the biliverdin cofactor correlates with the enzymatic readout of the effector domain.

Since light intensity is apparently not limiting the steady-state P$_{fr}$ population under red light, we denatured *Is*PadC directly after illumination with saturating red light to address the isomerization state of the biliverdin cofactor. To this end, we followed the protocol of denaturation under acidic conditions described by Thümmler *et al.* (*Thümmler et al., 1981*) and observed that more than 90% of the biliverdin population is present as the 15*E* isomer during constant red light illumination (*Figure 1e*). The comparison with *Dr*BphP clearly shows that similar levels of the isomerized biliverdin species are present upon red light activation, however, even complete isomerization of biliverdin apparently does not allow full population of the P$_{fr}$ state in the case of *Is*PadC (cf. *Figure 1c*). This is either due to different biliverdin environments of the two protomers forming an *Is*PadC dimer or due to a steady-state equilibrium of P$_{fr}$ with preceding intermediates along the activation pathway (*Borucki et al., 2005*; *Buhrke et al., 2018*; *Dasgupta et al., 2009*). By following the thermal reversion of 15*E* biliverdin to its 15*Z* isomer by means of quenching the protein after different dark adaptation times, mean lifetimes and amplitudes are observed that are comparable to those obtained for the native protein (inset of *Figure 1e*), which indicates a direct correlation of the native spectral properties with the isomerization state of biliverdin. Interestingly, at the high concentrations of 10 µM required to perform this experiment, fitting of the data required the sum of three exponentials (Table 1 in *Supplementary file 1*). While this fitting function and the obtained parameters differ from previously reported data for *Is*PadC (*Gourinchas et al., 2017*), we realized that these

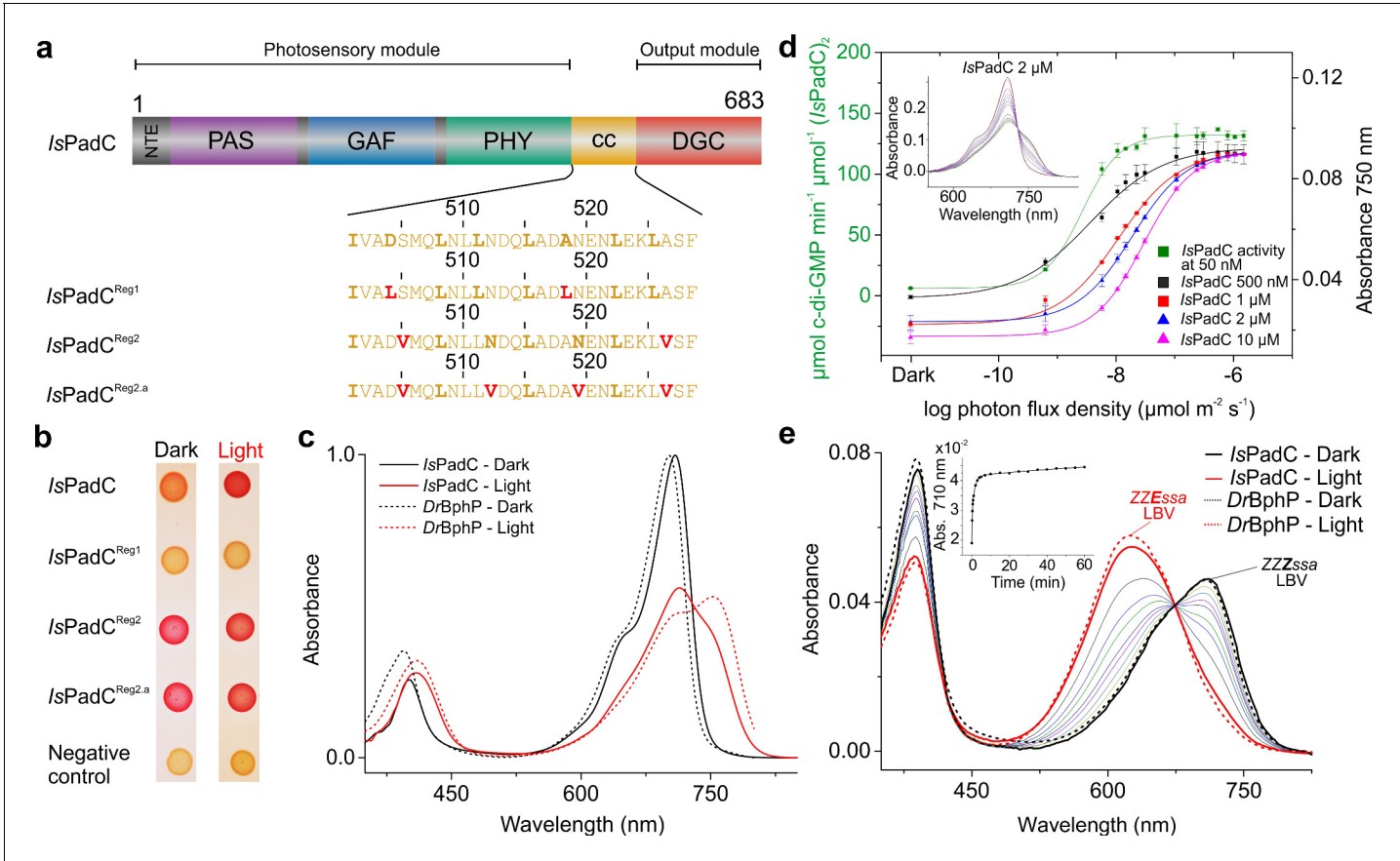

**Figure 1.** *Is*PadC constructs and their functional characterization. (**a**) Schematic representation of *Is*PadC and coiled-coil linker variants. Individual domains are colored in dark gray, violet, blue, green, orange, and red for the N-terminal extension (NTE), PAS, GAF, PHY, coiled-coil (cc), and DGC domains, respectively. The coiled-coil sequence is shown for each linker variant with red-colored residues indicating substituted amino acids. Bold letters refer to the hydrophobic *a* and *d* positions of the *abcdefg* heptad repeat units of the coiled-coil. Wild-type and *Is*PadC[Reg1] are expected to feature the inhibiting coiled coil register 1, whereas *Is*PadC[Reg2] and *Is*PadC[Reg2.a] are stabilized in a different coiled-coil register referred to as register 2 (stimulating). (**b**) In vivo screening of DGC activity. *Is*PadC shows the expected increased red coloration upon red light illumination activation (*Gourinchas et al., 2017*). By contrast the coiled coil variant stabilized in the inhibiting register (*Is*PadC[Reg1]) shows no activity as reflected by a white coloration, whereas the variants stabilized in the stimulating register (*Is*PadC[Reg2], *Is*PadC[Reg2.a]) show light-independent red coloration. (**c**) UV/Vis characterization of *Is*PadC showing the dark state spectrum in comparison to one obtained directly after red light illumination. For comparison, spectra acquired under identical light and measurement setting using *D. radiodurans* phytochrome (*Dr*BphP) are shown (**d**) Effect of red light photon flux density on enyzmatic activity and on the spectral contribution at 750 nm. To account for the different protein concentrations, all measurements were scaled based on the respective dark state absorption of the P_r peak. To account for reduced Pr contributions due to partial light activation by the measuring light at low protein concentrations, the 750 nm contributions were subsequently normalized to saturating red light conditions of the 2 μM sample. Full spectra at various flux densities are shown for the 2 μM *Is*PadC concentration in the inset. (**e**) Thermal recovery of 15*E* biliverdin followed by denaturation of 10 μM native samples at different time points. For comparison *Dr*BphP has been denatured in the dark and under illuminated conditions.

DOI: https://doi.org/10.7554/eLife.34815.002

The following figure supplements are available for figure 1:

**Figure supplement 1.** Coiled-coil linker sequences and heptad repeat assignments.

DOI: https://doi.org/10.7554/eLife.34815.003

**Figure supplement 2.** Characterization of sequence variants of the *Is*PadC coiled-coil linker.

DOI: https://doi.org/10.7554/eLife.34815.004

**Figure supplement 3.** Spectral characterization of *Is*PadC[Δ442-477::SG].

DOI: https://doi.org/10.7554/eLife.34815.005

**Figure supplement 4.** Kinetic characterization of product formation of *Is*PadC[Reg1] (a), *Is*PadC[Reg2] (b), and *Is*PadC[Reg2.a] (c), respectively.

DOI: https://doi.org/10.7554/eLife.34815.006

observations directly correlate with the protein concentration used for the different experiments. At low protein concentrations, fitting with the sum of two exponentials is sufficient to describe the recovery kinetics, however, increasing the protein concentration results in apparently slower recoveries until at some point the inclusion of a third exponential is indicated – eventually providing comparable values for the time constants of the first two phases (cf. Table 1 in *Supplementary file 1*). Importantly, also changing the salt concentration of the samples significantly impacts the obtained recovery kinetics. This, in combination with the effect of protein concentration, points towards an involvement of the protein oligomerization state and/or non-specific interactions of structural elements close to the chromophore binding site in modulating the recovery kinetics of the biliverdin cofactor.

The spectroscopic characterization of the three *Is*PadC register variants revealed no major differences compared to the wild type with respect to their dark- and light-state UV/Vis absorption spectra (*Figure 1—figure supplement 2a–c*). The absorption maxima of the $P_r$ states are shifted slightly from 710 nm for the wild-type to 708 nm for both *Is*PadC$^{Reg1}$ and *Is*PadC$^{Reg2.a}$ as well as to 707 nm for *Is*PadC$^{Reg2}$. Consequently, the difference spectra of light-dark (*Figure 1—figure supplement 2a–c*) show analogous changes in maxima of the negative $P_r$ signals, whereas the signals associated with $P_{fr}$ formation remain almost unaffected. The thermal recoveries to the ground state $P_r$ absorption spectra, however, are pronouncedly affected by the amino acid substitutions in the coiled-coil linker (Table 1 in *Supplementary file 1* and *Figure 1—figure supplement 2d*). We also observed that the protein concentration has a pronounced effect on the recovery kinetics of the register variants and have therefore decided to use 2 µM protein samples for the comparison of individual protein variants. Since this concentration differs from that used for the initial *Is*PadC characterization (*Gourinchas et al., 2017*) the reported values in Table 1 in *Supplementary file 1* also differ slightly from previously reported data for *Is*PadC (*Gourinchas et al., 2017*). Importantly, recovery kinetics of wild-type *Is*PadC and the *Is*PadC$^{Reg2}$ variant could be fitted with the sum of two exponentials, whereas *Is*PadC$^{Reg1}$ and *Is*PadC$^{Reg2.a}$ required the addition of a third exponential function to provide satisfying fits already at low protein concentrations. Interestingly, the latter variants are presumably extensively stabilized in their coiled-coil registers (inhibiting register 1 and stimulating register 2, respectively) and feature at least one phase with extremely slow recovery, whose contribution to the overall amplitude differs pronouncedly (*Figure 1—figure supplement 2d*). Especially *Is*PadC$^{Reg2.a}$ appears to be severely limited in fully recovering to the $P_r$ ground state, which shows that the conformation of the coiled-coil register and its dynamics strongly influence the biliverdin environment.

A closer inspection of the recovery kinetics by comparison of data measured at 710 nm (reformation of $P_r$) and 750 nm (depletion of $P_{fr}$ and potential non-$P_{fr}$ photoproducts) revealed that the time constants of individual recovery phases match between the 710 and 750 nm analyses. Interestingly, also the relative amplitudes contributing to individual phases at 710 and 750 nm appear to correlate for all constructs. While this observation might suggest that the recoveries do not involve photoproducts that significantly differ in their spectra from the $P_r$ or $P_{fr}$ absorbing species, it does not exclude the presence of closely related species such as for example meta-R states (*Borucki et al., 2005*; *Buhrke et al., 2018*; *Wagner et al., 2008*). The observation of a non-isosbestic recovery (*Figure 1—figure supplement 2e*) further supports the involvement of at least one colored species in addition to $P_r$ and $P_{fr}$. In fact, the fairly constant amplitude contributions of the fastest phase of the thermal recoveries (~36%, cf. Table 1 in *Supplementary file 1*) suggest that this phase might correspond to the recovery of a biliverdin species with lower extinction coefficient bound to one PadC protomer, whereas the remaining phase(s) reflect the recovery of the second biliverdin species – most likely the classical $P_{fr}$ state. A closer inspection of the thermal recoveries in the region of the expected isosbestic points between $P_r$ and $P_{fr}$ as well as $P_r$ and meta-R (cf. ref [*Borucki et al., 2005*]) confirmed that the relative amplitudes of the individual contributions change significantly in this spectral region and that the maximal contribution of the meta-R and $P_{fr}$ species lies close to the isosbestic point of the respective other contribution. Therefore, our initial spectroscopic characterization indicates that two different biliverdin species are stabilized upon red light illumination of *Is*PadC in the two protomers of the functional parallel phytochrome dimer.

Such an asymmetric activation of *Is*PadC and its variants is also supported by the observations made for a construct lacking the PHY-tongue element (*Is*PadC$^{Δ442-477::SG}$) (*Gourinchas et al., 2017*). The spectral characteristics of the illuminated *Is*PadC$^{Δ442-477::SG}$ construct are reminiscent of a composite of $P_r$ and meta-R states considering meta-R species characterized for D207 and H260 variants

of *Dr*BphP (*Wagner et al., 2008*) or low temperature trapped forms of Agp1 from *Agrobacterium tumefaciens* (*Borucki et al., 2005*). The meta-R contribution is also supported by the characteristic changes in the light-dark difference spectrum compared to the other *Is*PadC variants (*Figure 1—figure supplement 2f* and *Figure 1—figure supplement 3a*). Importantly, acidic denaturation of *Is*PadC$^{\Delta442-477::SG}$ revealed that in comparison to wild-type *Is*PadC only half of the biliverdin population can be isomerized to the 15*E* form (*Figure 1—figure supplement 3b*). The biliverdin cofactor bound to the respective other protomer of the dimeric species radiatively decays to the ground state rather than isomerizing, as shown by the significantly increased fluorescence of *Is*PadC$^{\Delta442-477::SG}$ (*Figure 1—figure supplement 3c*). Interestingly, the spectral properties of this partial light-activated state are retained over hours (*Figure 1—figure supplement 3a*) and even over-night incubation does not allow recovery to the P$_r$ ground state (*Gourinchas et al., 2017*).

These data are consistent with a mechanism of *Is*PadC light activation involving light-induced isomerization of both biliverdin molecules of dimeric *Is*PadC. However, only one biliverdin moiety can undergo all subsequent structural rearrangements to obtain a classical P$_{fr}$ state. The second biliverdin is most likely trapped in an early meta-R like conformation. This asymmetry is more pronounced for the register variants stabilized in register 2, where the P$_{fr}$ environment is significantly stabilized relative to the wild-type and the *Is*PadC$^{Reg1}$ variant. Considering that the stimulating register and activation of enzymatic activity by illumination are coupled in wild-type *Is*PadC, we also had a closer look at the effect of stabilizing the coiled-coil in the two registers on catalytic turnover of GTP to c-di-GMP.

## The kinetic characterization of *Is*PadC variants reveals a complex interplay of enzymatic activity and coiled-coil conformations

In order to quantify the light regulation capacities of individual *Is*PadC register variants we employed an HPLC-based assay that enables baseline separation of the substrate GTP, the pppGpG intermediate and the product c-di-GMP. None of the analyzed register variants showed significant levels of intermediate formation as observed for *Is*PadC homologs or some linker deletion variants characterized previously (*Gourinchas et al., 2017*). Similar to *Is*PadC the conversion of substrate to product tends to be inhibited by increasing substrate concentrations, however, this phenomenon is less pronounced for both variants designed to stabilize the stimulating register 2 (*Figure 1—figure supplement 4a–c*). *Table 1* summarizes the obtained rate constants for product formation at a substrate concentration of 200 µM, which corresponds to the maximum of GTP conversion for *Is*PadC$^{Reg2}$. At this concentration, no light regulation can be observed for *Is*PadC$^{Reg1}$ and *Is*PadC$^{Reg2.a}$, whereas *Is*PadC$^{Reg2}$ still shows residual stimulation of DGC activity by red light (roughly 2-fold). Considering that *Is*PadC$^{Reg2}$ activity can be increased whereas *Is*PadC$^{Reg2.a}$ cannot, indicates that a too strong stabilization of the coiled-coil in the stimulating register can have negative effects for c-di-GMP formation. Apparently, the coiled-coil conformation of register 2 has a positive effect on enzymatic activity, but also the conformational dynamics of this structural element influence overall enzymatic turnover of the *Is*PadC system.

**Table 1.** Comparison of PadC kinetics of substrate conversion

| Construct | Comparison of initial rates* @ 200 µM GTP (µmol product min$^{-1}$ µmol$^{-1}$ enzyme$_2$) | | |
|---|---|---|---|
| | Dark state | Light state | Fold activation |
| *Is*PadC (*Gourinchas et al., 2017*) | 0.8 ± 0.1 | 34.2 ± 0.4 | 44x |
| *Is*PadC$^{Reg1}$ | 2.13 ± 0.05 | 2.12 ± 0.01 | - |
| *Is*PadC$^{Reg2}$ | 52 ± 4 | 117 ± 5 | 2.2x |
| *Is*PadC$^{Reg2.a}$ | 53.2 ± 0.4 | 53 ± 3 | - |

*Comparison of product formation between the various constructs was performed for initial reaction rates at 200 µM GTP. Initial rates are quantified from experimental triplicates for three time points, and the sample standard deviation of individual points contributed to the error estimation of the linear fit that is used to calculate the initial rate of product formation. The SE of the estimate from the linear regression is used as error indicator.

DOI: https://doi.org/10.7554/eLife.34815.007

# The crystal structure of *Is*PadC$^{Reg2}$ features an asymmetric P$_{fr}$/P$_r$ heterodimer

Addressing the structural implications of the stimulating register 2 conformation was enabled by the crystallization of *Is*PadC$^{Reg2}$ under dark conditions. Solving the crystal structure by molecular replacement (MR) using data to 2.85 Å (Table 2 in *Supplementary file 1*) was only possible after splitting the dark-state wild-type *Is*PadC structure into individual subdomains, due to substantial structural rearrangements of the overall *Is*PadC$^{Reg2}$ structure compared to the wild-type (*Gourinchas et al., 2017*). Even though *Is*PadC$^{Reg2}$ and wild-type *Is*PadC form crystals with the same overall lattice, we observed interesting rearrangements in the DGC effector domains that are coupled to the transition of the coiled-coil into the expected register 2 conformation. In addition, one PHY domain and its associated tongue element differ substantially from the dark-state *Is*PadC structure described previously (*Gourinchas et al., 2017*).

In fact, already after the initial placement of all *Is*PadC domains by MR, the density of one protomers PHY-tongue extension did not match the β-hairpin structure characteristic for the dark-state P$_r$ environment of biliverdin. Rather the electron density supported an α-helical element that is usually only observed for P$_{fr}$-conformations (*Takala et al., 2014*; *Yang et al., 2009*; *Burgie and Vierstra, 2014*) (*Figure 2a–b* and *Figure 2—figure supplement 1*). A closer inspection of the cofactor binding site revealed that also biliverdin in its classical *ZZZssa* conformation would not fit into the observed electron density. However, models of biliverdin in its isomerized and rotated 15*E* conformation nicely explain the observed electron density as supported by the polder OMIT maps (*Liebschner et al., 2017*) of the co-factor region (*Figure 2—figure supplement 2*). The structural rearrangements of the biliverdin environment are, however, only observed for one protomer, whereas the other protomer features the expected dark-state P$_r$ environment observed for both protomers in the wild-type *Is*PadC structure (*Gourinchas et al., 2017*). Therefore, the *Is*PadC$^{Reg2}$ structure features an asymmetric dimer (*Figure 2a*) that is composed of one P$_{fr}$ biliverdin environment (protomer A) and one P$_r$ environment (protomer B). In addition to the helical tongue and the *ZZEssa* biliverdin configuration, also the side chains of residues Y168, H193 and F195 in the proximity of the rotated D-ring change their rotamer positions as described for other phytochrome P$_{fr}$ states (*Takala et al., 2014*; *Yang et al., 2009*; *Burgie and Vierstra, 2014*) (*Figure 2b* and *Figure 2—figure supplement 2*).

The P$_{fr}$/P$_r$ heterodimer of the final *Is*PadC$^{Reg2}$ model is further supported by UV/Vis absorption spectra acquired for *Is*PadC$^{Reg2}$ crystals in comparison to wild-type *Is*PadC crystals. Averaging multiple spectra collected at different orientations of crystals relative to the incident measuring light, revealed a clear shoulder around 750 nm for the *Is*PadC$^{Reg2}$ crystal spectra (*Figure 2—figure supplement 3*). While a quantitative comparison to wild-type *Is*PadC crystal spectra is complicated by the preferred orientation of molecules in the crystal lattice and the resulting effect on extinction coefficients, the almost identical crystal lattices and morphologies of the two crystals enable a qualitative comparison of their spectra after baseline subtraction and scaling to their Soret band absorption. The comparison of both crystal spectra clearly demonstrates that the *Is*PadC$^{Reg2}$ crystals contain a mixture of P$_r$ and P$_{fr}$ populations that reflects the refined P$_{fr}$/P$_r$ heterodimer.

The asymmetry of the *Is*PadC$^{Reg2}$ structure extends from the biliverdin environment and the associated PHY-tongue to the PHY domain itself. Refolding of the tongue hairpin provides the structural flexibility for the PHY domain of chain A to reposition relative to chain B resulting in pronounced rearrangements at the PHY dimer interface. This repositioning of the PHY dimer correlates with the structural rearrangements of the coiled-coil to populate the register 2 conformation. In fact, the terminal PHY-helix directly extends into the coiled-coil sensor-effector linker and the whole structural element undergoes a hinge-like motion that is related to the structural transitions observed for various parallel phytochrome dimers (*Gourinchas et al., 2017*) (*Figure 2c*). Considering that more than 40 residues are part of this extended helical region, the global structural rearrangements are a complex mixture of individual sub-regions. Especially for the central part of the coiled-coil, the hinge-like separation of the corresponding residues is not very pronounced (*Figure 3* and *Figure 3—figure supplement 1*) and, due to the absence of any rotational contribution to the transition between registers 1 and 2, is approximated as a quasi-translational movement throughout the manuscript. Interestingly, the hinge-like rearrangement at the central helical spine of the phytochrome module breaks the Arg327-Asp494 salt bridge at the PHY interface. This places the sidechains of Arg327 of

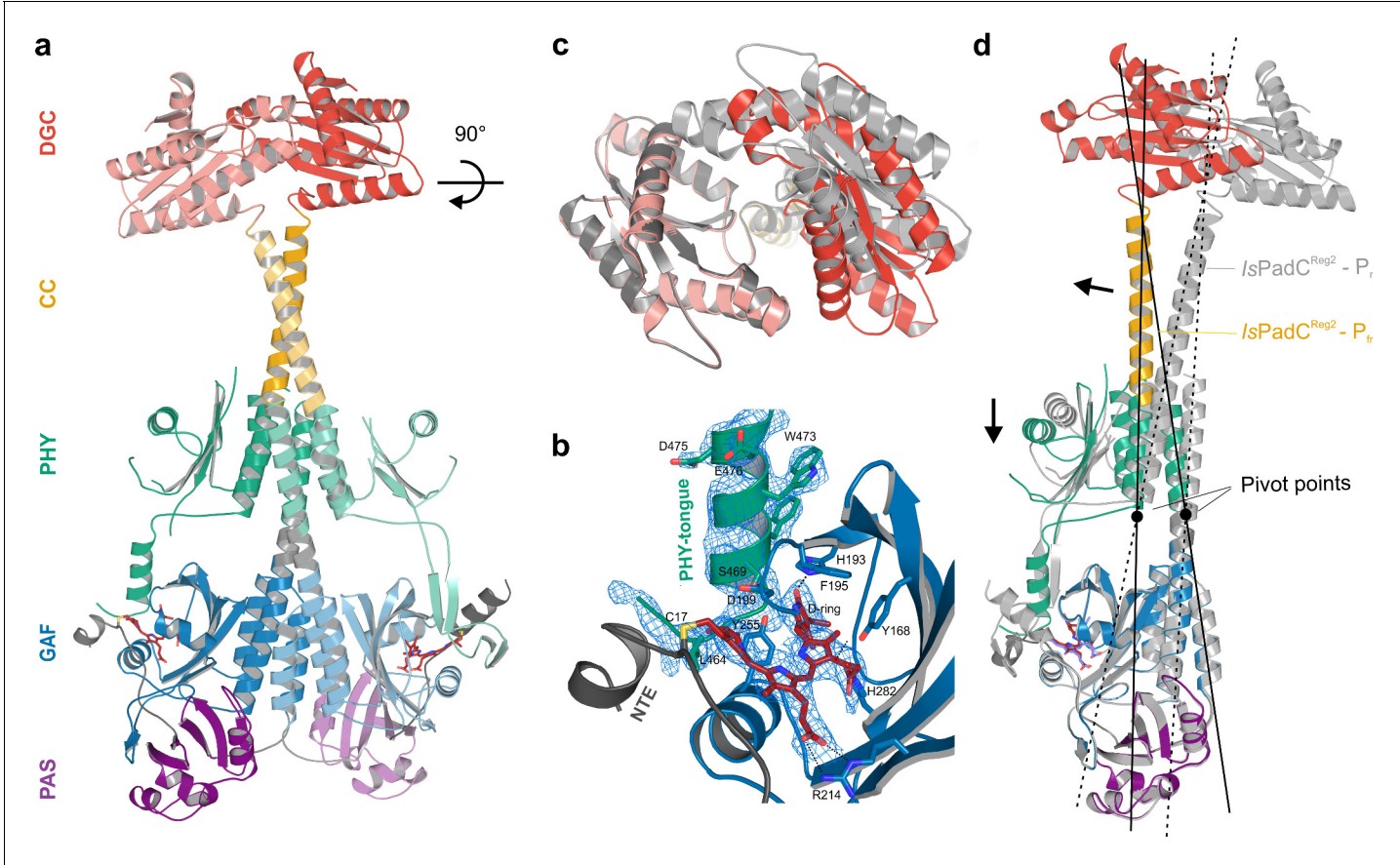

**Figure 2.** Asymmetric activation of a bacteriophytochrome dimer. (**a**) Crystal structure of $IsPadC^{Reg2}$ in cartoon representation with individual domains colored according to *Figure 1a*. Within the dimer, the activated protomer is highlighted in bold colors. The biliverdin chromophore is colored in dark red and represented in sticks. (**b**) Close-up view of the biliverdin-binding pocket corresponding to the $P_{fr}$ conformation, highlighting the secondary structure change of the PHY-tongue element. The $2F_o$-$F_c$ electron density map contoured at $1\sigma$ around the chromophore is shown as light blue mesh. Important residues are shown in sticks. For clarity residues 200–213 have been hidden from the view. (**c**) Top view of the asymmetric GGDEF dimer induced by the coiled-coil linker helix translation. The GGDEF dimers are superimposed by aligning protomer A of $IsPadC^{Reg2}$ to the corresponding protomer B (residues 529–683) and the moved copy of the GGDEF dimer is represented as gray cartoon. (**d**) Superposition of the $P_{fr}$ protomer of $IsPadC^{Reg2}$ (colored as in **a**) and the $P_r$ protomer of $IsPadC^{Reg2}$ (colored in light gray) based on the PAS-GAF core (residues 1–312) of the respective chain A and B. Arrows indicate the structural rearrangements associated with $P_{fr}$ formation. The pivot points and the associated lines show the hinge-like rearrangements at the central helical spine and the terminal PHY-helix extending into the coiled-coil linker.

DOI: https://doi.org/10.7554/eLife.34815.008

The following figure supplements are available for figure 2:

**Figure supplement 1.** $P_r$ and $P_{fr}$ models and electron density of the PHY-tongue regions of $IsPadC^{Reg2}$.

DOI: https://doi.org/10.7554/eLife.34815.009

**Figure supplement 2.** Polder-maps generated for the cofactor binding site with chromophores omitted from both chains of $IsPadC^{Reg2}$ and wild-type $IsPadC$.

DOI: https://doi.org/10.7554/eLife.34815.010

**Figure supplement 3.** Crystal spectra of $IsPadC^{Reg2}$ and $IsPadC$ at 100K.

DOI: https://doi.org/10.7554/eLife.34815.011

both protomers in close proximity potentially contributing to the dynamic nature of the PHY interface upon illumination (*Figure 3—figure supplement 1c*).

The quasi-translational reorientation of the coiled-coil register extends to the very end of the sensor-effector linker, which is directly linked to the strictly conserved DXLT motif of DGCs (*Römling et al., 2017*). The structural consequences of the coiled-coil rearrangement are a different spacing of the two structural elements corresponding to the functionally important DXLT sequence in the dimeric assembly. Since these structural elements are directly linked to the open active site at

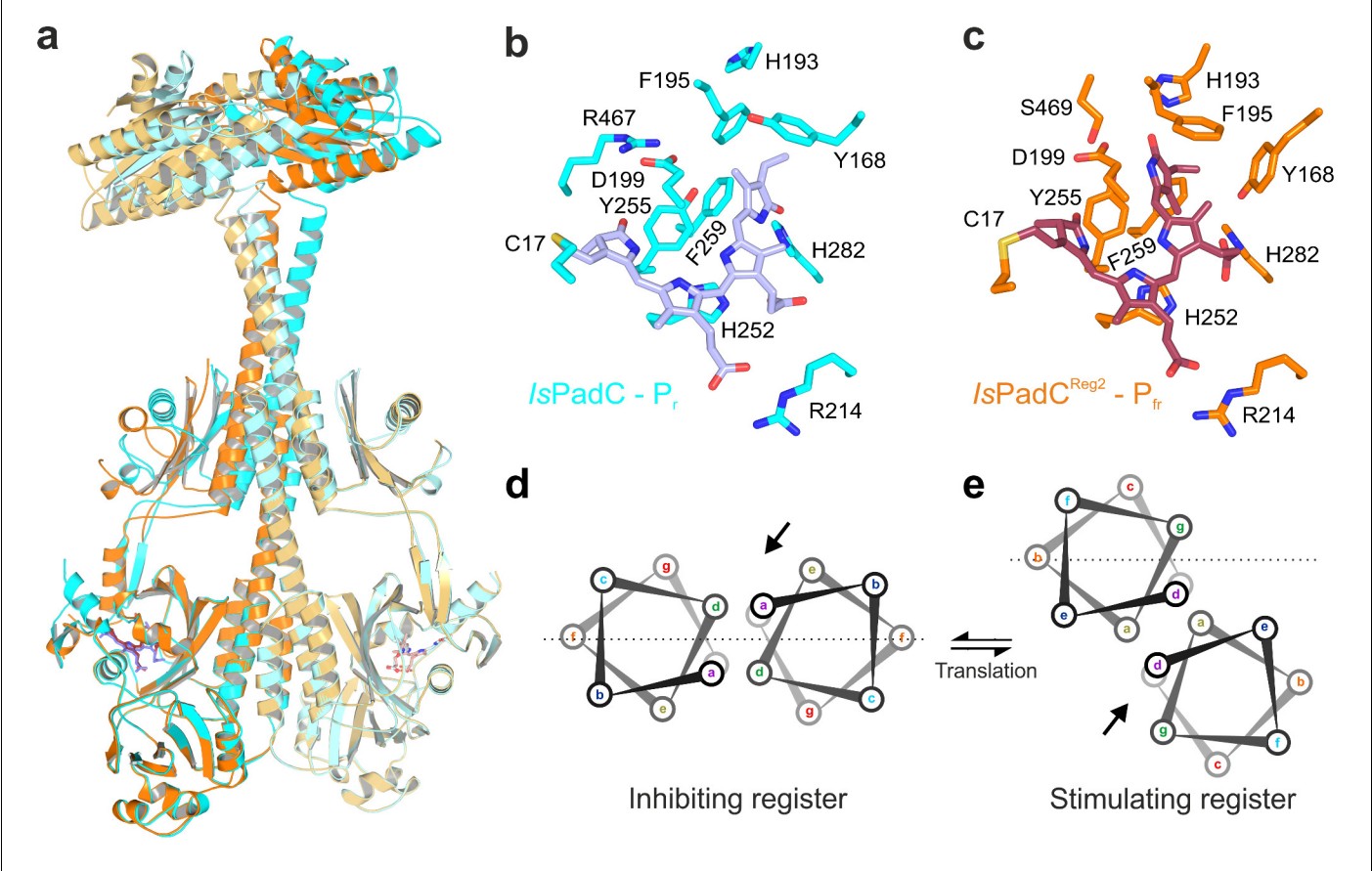

**Figure 3.** Structural rearrangements in the dimer conformation upon PHY-tongue refolding. (a) Superposition of the *Is*PadC dark-state structure (colored in cyan) with the activated *Is*PadC$^{Reg2}$ structure (colored in orange) based on the PAS-GAF cores (residues 1–312) of the respective chains B. The superposition shows the quasi-translational rearrangement in the middle of the coiled-coil linker associated with PHY-tongue refolding of one protomer. The superposition of the $P_r$-state phytochrome protomers of *Is*PadC and *Is*PadC$^{Reg2}$ (respective chains B) is almost identical (RMSD of 0.26 Å over 347 Cα-atoms). (b–c) Stick representation of residues around the biliverdin D-ring for the *Is*PadC $P_r$-state (b) and the *Is*PadC$^{Reg2}$ $P_{fr}$-state (c), respectively. The *ZZEssa* configuration of biliverdin results in rotamer repositioning of Tyr168, Phe195, His193, Tyr255, and results in loss of the Asp199-Arg467 salt bridge between the GAF domain and the PHY-tongue. (d–e) The stimulating register 2 conformation of the coiled-coil linker is populated by translation of one linker helix upon PHY-tongue refolding.

DOI: https://doi.org/10.7554/eLife.34815.012

The following figure supplements are available for figure 3:

**Figure supplement 1.** Flexibility of the coiled-coil linker element in response to the molecular cross-talk between sensor and effector.
DOI: https://doi.org/10.7554/eLife.34815.013

**Figure supplement 2.** Allosteric effects of the coiled-coil rearrangement on the conformation of the DGC dimer.
DOI: https://doi.org/10.7554/eLife.34815.014

the interface of the two GGDEF domains, the coiled-coil register switching apparently has direct implications for DGC activity. In this context, it should be emphasized that the *Is*PadC$^{Reg2}$ structure has been solved in almost the same crystal lattice as the wild-type *Is*PadC dark state structure. This implies that the conformational rearrangements of the DGC dimer are associated with the coiled-coil register transition and the asymmetric activation of the sensory module rather than being affected by lattice specific crystal contacts. Interestingly, superimposing the DGC dimer onto itself by aligning one protomer with the corresponding other protomer (residues 529–683 chain A over chain B) reveals a pronounced asymmetry of the DGC dimer interface in *Is*PadC$^{Reg2}$ (*Figure 2c*), which significantly differs from the DGC dimer interface of the *Is*PadC dark state (RMSD 0.370 over 141 α-carbon and RMSD 0.267 over 138 α-carbon of chain A superimposed over chain B for *Is*PadCReg2 and *Is*PadC, respectively). Overall we observe a rotational movement of the DGC

protomer linked to the repositioned coiled-coil linker helix. This leads to an opening of the DGC dimer at one side and a closing at the other side (*Figure 2c* and *Figure 3—figure supplement 2a–b*).

In the absence of substrate or product bound to the *Is*PadC$^{Reg2}$ structure it is, however, not possible to associate a specific functional state to this DGC architecture. Nevertheless, the fact that *Is*PadC$^{Reg2}$ features constitutively high GTP turnover suggests that the conformational opening of the DGC dimer might either increase the affinity for GTP and/or bring two GTP moieties after binding into an energetically favorable position to catalyze the first phosphodiester bond formation. Interestingly, the DGC dimer architecture does not resemble any deposited GGDEF dimer in an inhibited conformation (*Chan et al., 2004*; *Wassmann et al., 2007*; *De et al., 2008*; *De et al., 2009*; *Navarro et al., 2009*), suggesting that the observed dimer might correspond to a functionally important conformation along the catalytic cycle of DGCs. In fact, the relative position of the two catalytic GGEEF sites at the center of the DGC dimer is affected by the asymmetric rearrangements in *Is*PadC$^{Reg2}$ and the net effect is a shortened distance between the GGEEF sites potentially facilitating the contact between the C3 ribose hydroxyl group of one GTP and the alpha phosphate group of the respective other GTP molecule (*Chan et al., 2004*; *Schirmer and Jenal, 2009*) (*Figure 3—figure supplement 2c*). This could provide the geometry required for the initial phosphodiester bond formation step, generating the pppGpG intermediate. Subsequent structural rearrangements would then be required to complete the second phosphodiester bond formation step and ultimately the release of the c-di-GMP product. That the observed architecture corresponds to one functionally important state along this pathway is further supported by the observed asymmetry of specific conserved residues close to the binding site of the GTP base (e.g. Lys569 and Tyr670) that change their rotamer positions and relative orientation to the other protomer (*Figure 3—figure supplement 2d*). In fact, these structural rearrangements are similar to changes induced by substrate binding as observed in the GTP soaked wild-type structure (*Gourinchas et al., 2017*).

Considering the importance of dynamic rearrangements during the catalytic cycle of converting two molecules of GTP to c-di-GMP via the pppGpG intermediate, it is unlikely that a single conformation observed in a crystal structure will correspond to 'the active' conformation of DGCs. To further address the importance of conformational dynamics on DGC activity and on the coupling of the coiled-coil region with both the enzymatic effector and the red-light sensor, we performed hydrogen-deuterium exchange experiments analyzed by mass spectrometry (HDX-MS).

## The importance of conformational dynamics for the regulation of DGC activity

Extending our previous HDX-MS characterization of wild-type *Is*PadC (*Gourinchas et al., 2017*), we performed a complementary analysis with the *Is*PadC$^{Reg2}$ system, which is almost constitutively active but still shows residual light regulation of enzymatic activity. Comparing just the differences in deuterium incorporation between light- and dark-state conditions for the two systems reveals no major differences. For both the wild-type and the *Is*PadC$^{Reg2}$ variant we observed an increase in conformational dynamics in the PHY-tongue region, the central helical spine and the coiled-coil sensor–effector linker upon illumination (*Figure 4*). This increase in conformational dynamics, especially for the coiled-coil linker, has recently been implicated in increasing DGC activity by providing the conformational flexibility to switch between the two coiled-coil registers (*Gourinchas et al., 2017*). A closer inspection of the deuterium exchange kinetics, however, revealed that the conformational dynamics of *Is*PadC$^{Reg2}$, compared to the wild-type protein in either the dark-adapted state or during constant red light illumination, are reduced in several functionally important regions. This is especially true for the PHY-tongue region, the coiled-coil and a GGDEF element close to the GTP binding site (*Figure 4* and *Figure 4—figure supplement 1*). The stabilization observed at the DGC dimer interface directly reflects the molecular structure of *Is*PadC$^{Reg2}$ with respect to the close interactions between the α0 helices and the DXLT motifs as well as the interaction of the α1/α2 and α4 helices on the closed side of the DGC interface. Obviously, the general stabilization of the GGDEF dimer also affects the conformational flexibility of the GGEEF motif.

Of special interest are the reduced conformational dynamics of the coiled-coil, since we previously correlated an increase in conformational dynamics with an increase in activity. Because *Is*PadC$^{Reg2}$ is more active than the wild-type in the dark, the reduced dynamics observed for this variant

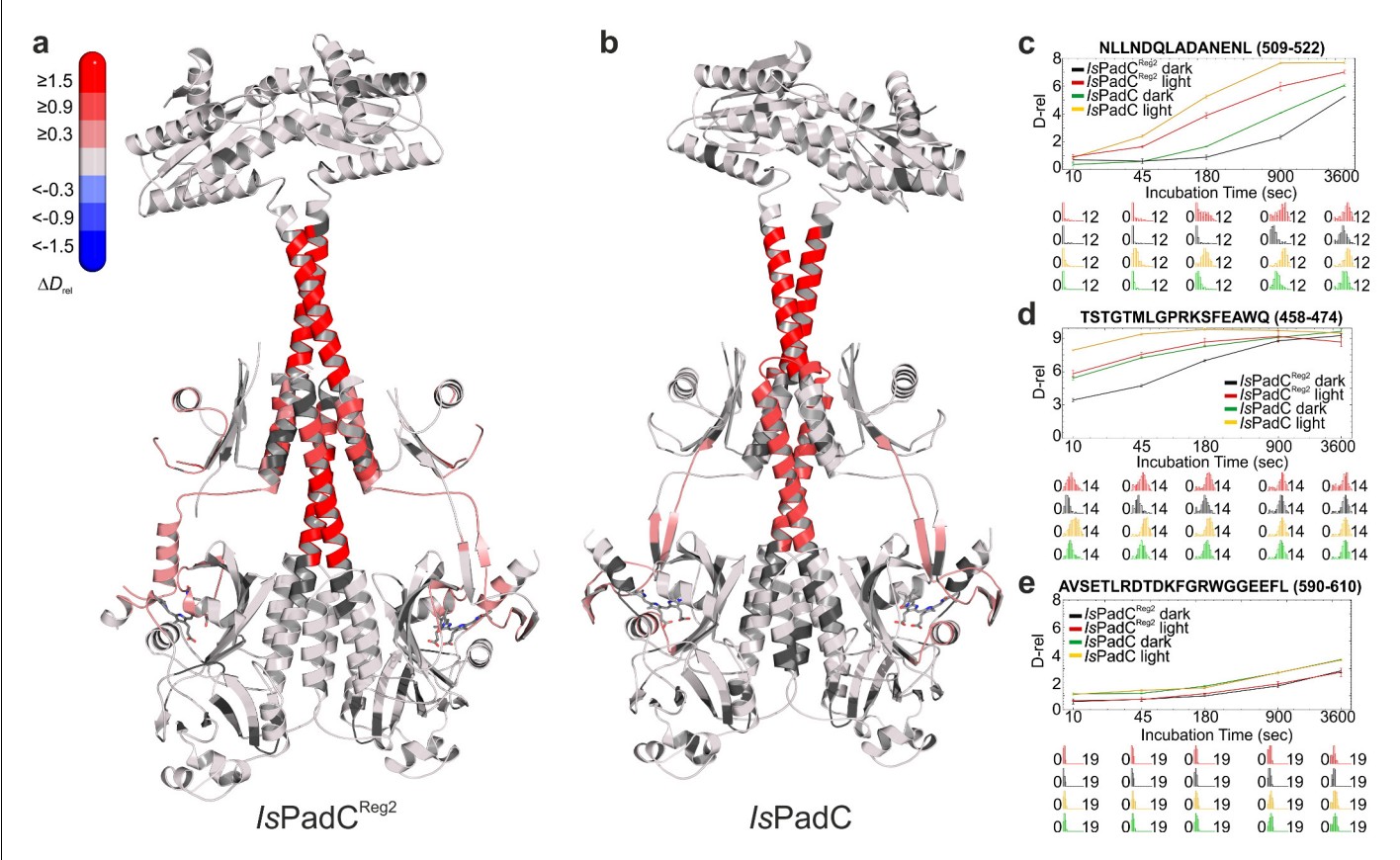

**Figure 4.** Comparison of the changes in conformational dynamics upon illumination of *Is*PadC^Reg2 and *Is*PadC. (a–b) Changes in conformational dynamics upon illumination of *Is*PadC^Reg2 (a) and *Is*PadC (b), respectively, evaluated by HDX-MS. Since time dependent deuterium uptake at the amide positions for different peptides correlates with the stabilization of both β-sheets and α-helices, the deuterium uptake of individual peptides can be related to the stability of the secondary structure elements in a defined region. Closer inspection of individual peptides (panels c–e), revealed that the absolute conformational dynamics of individual peptides differ substantially. The presented structures are colored according to the observed changes in relative deuterium incorporation ($\Delta D_{rel}$) between light-state and dark-adapted state ($D_{rel}$ light – $D_{rel}$ dark) after 15 min of deuteration. The changes in $\Delta D_{rel}$ are indicated by the scale in the top left corner with blue corresponding to reduced deuterium incorporation and red reflecting increased exchange of amide protons upon red light illumination. The biliverdin chromophore is represented as sticks and colored in gray. (c–e) Comparison of deuterium uptake curves of *Is*PadC^Reg2 and *Is*PadC peptides in the coiled-coil linker region (c), in the PHY-tongue region (d), and in the GGEEF element of the DGC (e), respectively. $D_{rel}$ is plotted against the deuteration time for light- and dark-state HDX-MS experiments. The error indicators correspond to the sample standard deviation of triplicate measurements. The lower parts show software-estimated abundance distributions of individual deuterated species on a scale from undeuterated to all exchangeable amides deuterated.

DOI: https://doi.org/10.7554/eLife.34815.015

The following figure supplements are available for figure 4:

**Figure supplement 1.** Comparison of changes in conformational dynamics of *Is*PadC^Reg2 and *Is*PadC evaluated by HDX-MS.
DOI: https://doi.org/10.7554/eLife.34815.016

**Figure supplement 2.** Individual deuterium incorporation plots of all evaluated peptides and comparison of common *Is*PadC^Reg2 and *Is*PadC peptides.
DOI: https://doi.org/10.7554/eLife.34815.017

suggest that it already populates a stable register 2 conformation in solution. The increase in deuterium incorporation upon illumination therefore only increases the dynamics of the system rather than switching to a specific register. Since *Is*PadC^Reg2 features a classical P_r state in solution, the structural rearrangements in the coiled-coil can apparently occur uncoupled from structural rearrangements in the biliverdin environment, for example linked to P_fr formation as observed in the crystal structure. These findings support the proposed switching mechanism between the two coiled-coil registers and show that both a structural component and a dynamic contribution are responsible for regulating enzymatic activity of the output module (**Figure 5**).

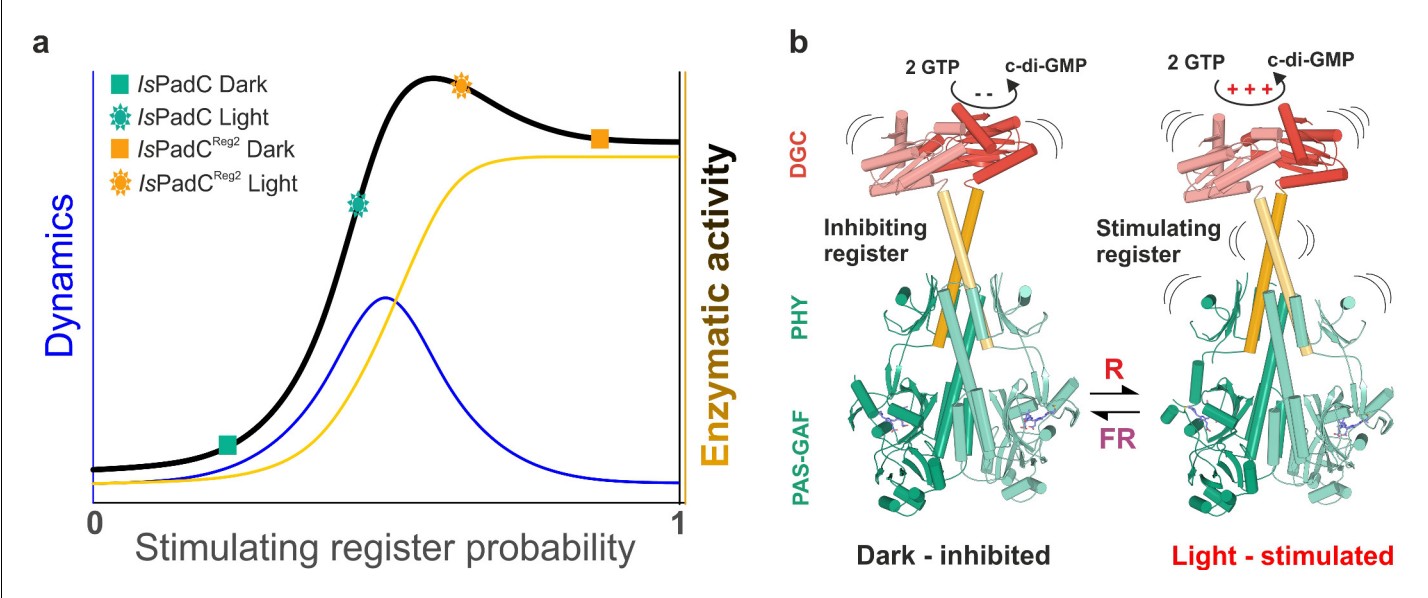

**Figure 5.** A toggle mechanism between linker conformations modulates DGC activity. (a) Schematic representation of how DGC activity is modulated by the coiled-coil linker register and its intrinsic conformational dynamics. The transition between the inhibiting and the stimulating coiled-coil conformations is represented on the x-axis as the probability to be in the stimulating register. Under conditions where a switching between coiled-coil registers is possible the dynamics of the system are increased (blue curve). In combination with the structural contribution of the stimulating register for DGC activity (orange curve), the overall enzymatic activity is approximated as the sum of both contributions (dark curve). A large photodynamic range can only be obtained by a fine-tuned balance of conformational flexibility and structural differences of the coiled coil conformations. (b) Schematic model of PadC activation. In the dark, a stable β-hairpin conformation of the PHY tongue elements maintains the coiled-coil linker in its inhibiting conformational register. Upon red light illumination and isomerization of the biliverdin of one protomer, the dimeric interface of the PSM rearranges leading to more conformational freedom at the PHY interface allowing the population of the stimulating linker register that in turn facilitates GTP conversion at the GGDEF domains.

DOI: https://doi.org/10.7554/eLife.34815.018

As far as the proposed asymmetry of *Is*PadC upon illumination is concerned, the HDX-data cannot give a decisive answer. However, it is interesting to note that several peptides at the central helical spine, the PHY-tongue and the coiled-coil feature broad isotope distributions during deuterium incorporation (*Figure 4—figure supplement 2*). These broad distributions are indications for bimodal deuterium uptake that originates from two molecular species with different deuterium exchange kinetics, which would be in line with two biliverdin environments coupled to different PHY-tongue conformations under steady-state light conditions.

## Discussion

### Negative cooperativity in signal integration upon chromophore isomerization

In the *Is*PadC (*Gourinchas et al., 2017*) system (*Figure 1a*), the bacteriophytochrome module is linked to a diguanylyl cyclase (DGC) effector (*Ryjenkov et al., 2005*) and controls formation of the bacterial second messenger cyclic-dimeric-GMP (*Römling et al., 2013*) by red light (*Figure 1a–b*). A closer inspection of illumination properties of *Is*PadC revealed red light activation already at low photon flux densities, but also showed only partial $P_{fr}$ formation at saturating light intensities (*Figure 1c–d*). Consequently, light-activated absorption spectra differ significantly from those of systems featuring highly stable $P_{fr}$ states (*Yang et al., 2009*; *Burgie et al., 2016*). Nevertheless, acidic denaturation at saturating light conditions revealed more than 90% of the biliverdin to be isomerized to the 15$E$ state (*Figure 1e*). In comparison to the *D. radiodurans* phytochrome (*Dr*Bphp) dimer that features two isomerized chromophores under saturating light (*Figure 1e*) and allows both phytochrome-tongue (PHY-tongue) elements to be refolded (*Takala et al., 2014*; *Burgie et al., 2016*), we

hypothesized that differences in the local environment of the bilin cofactors or at the dimeric interface of $Is$PadC destabilize the establishment of $P_{fr}$ environments on both protomers. Therefore, a fraction of chromophores, albeit being isomerized, might be trapped in a global $P_r$-like phytochrome environment that does not allow refolding of the tongue to populate the $P_{fr}$ state. Based on the spectral characteristics of $Is$PadC in comparison to several variants of $Dr$BphP (*Wagner et al., 2008*) and the importance of $P_{fr}/P_r$ heterodimers in plant phytochrome-B signaling (*Klose et al., 2015*) the local structural environment of the biliverdin cofactor in the non-$P_{fr}$ protomer most likely reflects a meta-R like state (*Borucki et al., 2005*) where side chains close to the biliverdin adapt their rotamer positions, however no tongue refolding can take place.

In general, excitation of the biliverdin chromophore (*Mroginski et al., 2007*; *Toh et al., 2010*) is followed by fast isomerization and slower local structural changes around the chromophore binding pocket that are propagated to the effector domains in full-length proteins (*Gourinchas et al., 2017*; *Björling et al., 2016*). Refolding of the PHY-tongue region from a β-hairpin to an α-helical structure (*Takala et al., 2014*; *Yang et al., 2009*; *Anders et al., 2013*; *Burgie et al., 2016*) also results in rearrangements at the phytochrome dimer interface and highlights the cross-talk between biliverdin activation and conformational dynamics of the dimeric assembly. Along the line of the relevance of the dimer interface, disruption of the PAS-GAF bidomain dimerization has been shown to impact thermal reversion of $P_{fr}$ in the case of $Dr$BphP (*Takala et al., 2015*). Similarly, allosteric effects of the dimer interface were shown to influence the mode of biliverdin interaction in engineered fluorescent proteins (*Stepanenko et al., 2016*). The influence of the dimeric interface on properties of the two biliverdin cofactors is further supported by the stabilization of the coiled-coil sensor-effector linker in different constructs of $Is$PadC. Following thermal relaxation of the variants revealed a pronounced effect of stabilization of register 2 on the recovery of the $P_{fr}$ contribution. At the same time, the fast reverting meta-R component is only slightly influenced by the conformational dynamics at the dimer interface. Most likely, the molecular mechanism of meta-R recovery follows a shunt-pathway similar to that described for *Agrobacterium fabrum*, Agp1 and Agp2 (*Buhrke et al., 2018*). The different effects of the dimer interface on the recovery of the different biliverdin species are an additional indication for the structural asymmetry of light-activated $Is$PadC. In fact, the correlation of increased $P_{fr}$ lifetimes with increased stabilization of the register 2 conformation of the linker at the same time as sustained prevention of $P_{fr}$ formation on the second protomer, as indicated by constant amplitudes for the recovery of the meta-R component (Table 1 in *Supplementary file 1*), shows that the biliverdin environments are functionally coupled via cooperative effects at the dimeric assembly (*Klose et al., 2015*). This special case of negative cooperativity indicates that $P_{fr}$ associated conformational changes in one protomer break the global symmetry of the dimer and result in an asymmetric assembly that prevents reorganization of the second biliverdin binding site and/or associated functional elements such as the PHY-tongue region.

Further support for the stabilization of heterodimers via the dimer interface originates from the characterization of an $Is$PadC$^{\Delta442-477::SG}$ variant (*Gourinchas et al., 2017*) in which the PHY-tongues have been replaced by a short loop. The increased fluorescence and the fact that only half of the biliverdin population can be isomerized to the 15$E$ state (*Figure 1—figure supplement 3b–c*) support the importance of the PHY-tongue elements for stabilization of the biliverdin environments. At the same time, the observed asymmetry in the $Is$PadC$^{\Delta442-477::SG}$ construct shows that structural changes at the dimer interface can stabilize different chromophore environments upon light activation of $Is$PadC; based on the reduced amplitude and the blue-shifted maximum of the light-dark difference spectrum (*Figure 1—figure supplement 3a*), we assume a composite of $P_r$ and meta-R states to be present upon light activation of $Is$PadC$^{\Delta442-477::SG}$.

## Structural rearrangements at the coiled-coil sensor-effector linker modulate DGC activity

In $Is$PadC, structural rearrangements at the dimer interface modulate the conformation of the coiled-coil sensor-effector linker and thereby regulate DGC activity (*Gourinchas et al., 2017*). More precisely, the coiled-coil sequence of the linker region adopts two conformational registers leading to either DGC activation or inhibition (*Figure 1b*, and *Figure 1—figure supplement 1*). In fact, the stabilization of the coiled-coil in its inhibited register ($Is$PadC$^{Reg1}$) led to low, light-independent basal activity comparable to the $Is$PadC dark state. By contrast, stabilizing the stimulating register ($Is$PadC$^{Reg2}$) led to a highly active variant with a residual light-induced increase in enzymatic activity. The

detailed characterization of variants of *Is*PadC featuring different degrees of stabilization of the inhibiting or stimulating linker registers revealed significant differences in functional regulation of the enzymatic effector (*Figure 1b*, and *Table 1*). In combination with the effects of the coiled-coil variants on properties of the photocycle described above, their direct influence on the enzymatic functionality of the output module highlights the central role of the coiled coil linker in both integrating the light signal and regulating DGC activity. Interestingly, the *Is*PadC$^{Reg2}$ variant features a high dark state activity and can be further stimulated by red light illumination, whereas *Is*PadC$^{Reg2.a}$, for which two conserved Asn residues in the coiled-coil element were exchanged by hydrophobic valines, is constitutively active and can no longer be affected by illumination (*Figure 1b*, and *Table 1*). The observed effect of substituting the Asn residues supports their functional relevance at central positions of the heptad units (*Gourinchas et al., 2017*; *Fletcher et al., 2017*) and indicates that a conformational switching between coiled-coil registers is required for optimal activation of DGC activity (*Figure 5*).

## An asymmetric $P_{fr}/P_r$ heterodimer crystal structure reveals the coupling of the stimulating coiled-coil register with $P_{fr}$ state formation

The crystal structure of the *Is*PadC$^{Reg2}$ variant stabilized in the stimulating register shows that the structural rearrangements of the coiled-coil transition between the two registers, as required for signal integration (*Gourinchas et al., 2017*), are compatible with $P_{fr}$ state formation in a single protomer. In combination with the spectral signature of a $P_{fr}$/meta-R heterodimer observed upon illumination of all *Is*PadC constructs, this further supports the possibility of an *Is*PadC heterodimer as a central functional species (*Figure 1d*). Upon isomerization and D-ring rotation of one biliverdin the conformational changes induced at the dimer interface might prevent structural reorganization of the PHY-tongue region close to the second biliverdin binding site. Depending on the degree of coupling between various functional elements, different equilibria of $P_r$, lumi-R, meta-R and $P_{fr}$ states can influence the ultimate structural rearrangements in the phytochrome systems. Based on the assignments of meta-R transitions that involve structural rearrangement of the tongue region in other phytochrome systems (*Björling et al., 2016*; *Stojković et al., 2014*; *Anders et al., 2014*), we suspect that an early meta-R like state is enriched during steady state light illumination in *Is*PadC. The different stabilities of $P_{fr}$ states in various dimeric phytochromes might therefore have evolved to optimize specific action spectra as well as to enable the formation of functional heterodimers. Eventually, the possibility to tune the relative amounts of various biliverdin environments by functionally important structural elements might also be involved in the adaptability of the sensory module to different output domains.

The influence of the dimeric interface on the spectral photoisomerization properties has previously also been observed in inhibited photoconversion of the dimeric chromophore-binding domain of *D. radiodurans* phytochrome that can partially be reestablished upon monomerization (*Auldridge et al., 2012*). In fact, including the PHY domain in *D. radiodurans* phytochrome PSM constructs or addressing the full-length system results in full conversion to a $P_{fr}/P_{fr}$ homodimer (*Takala et al., 2015*; *Auldridge et al., 2012*). Apparently, nature employs both $P_{fr}$/(meta-R or $P_r$) heterodimers and $P_{fr}/P_{fr}$ homodimers as functional units for regulating downstream effectors. This appears to be true for bacterial phytochromes as well as evolutionary distant plant phytochromes, where asymmetric properties on the level of different protomers as well as different functional states of individual phytochromes are being employed to regulate complex cellular processes (*Clack et al., 2009*; *Liu and Sharrock, 2013*; *Klose et al., 2015*; *Hennig and Schäfer, 2001*; *Sharrock and Clack, 2004*). However, when considering other organisms, the description of functional asymmetry in photoreceptor oligomers is rather scarce (*Heintz and Schlichting, 2016*; *Winkler et al., 2013*), although it apparently represents an interesting evolutionary approach to modulate photochromic behavior and signaling diversities. In general, asymmetry is emerging as a critical aspect of signal integration (*Neiditch et al., 2006*; *Narayanan et al., 2014*; *Chervitz and Falke, 1996*) as well as regulation of output functionalities (*Kim et al., 2017*; *Hu et al., 2013*). Based on spectral data provided for several phytochrome systems, functional activation through $P_{fr}$ state formation in a single protomer might be a prevalent mechanism conserved in the family of phytochrome-like sensors that has so far been underestimated (*Tarutina et al., 2006*; *Yang et al., 2007*; *2015*; *Loughlin et al., 2016*; *Essen et al., 2008*).

## Detailed comparison of the wild-type and *Is*PadC$^{Reg2}$ crystal structures

Interestingly, both the wild-type and the *Is*PadC$^{Reg2}$ crystal structures have been obtained from crystals grown under constant dark conditions. Even though both constructs display very similar P$_r$ state spectra in solution (*Figure 1c* and ref [*Gourinchas et al., 2017*].), *Is*PadC$^{Reg2}$ always features a small shoulder around 750 nm indicating a subpopulation of species present in the P$_{fr}$ conformation. Apparently the crystallization conditions favor the assembly of asymmetric P$_{fr}$/P$_r$ heterodimers existing in equilibrium with the predominant P$_r$/P$_r$ homodimer in solution. These heterodimers feature the coiled-coil in its stimulating register, which stabilizes one protomer in the P$_r$ and the other in the P$_{fr}$ conformation (*Figure 2* and *Figure 3*). The enrichment of the P$_{fr}$ species only occurs during crystal growth and not in solution, confirming that the amino acid substitutions in *Is*PadC$^{Reg2}$ alone are not responsible for the stabilization of P$_{fr}$ under dark conditions, as observed in bathyphytochromes (*Zienicke et al., 2013*). Nevertheless, the P$_{fr}$/P$_r$ heterodimer should not be considered a crystallization artifact, but rather reflects one conformational assembly of the full-length system that is energetically compatible with the stabilization of the coiled-coil in its stimulating register.

The pronounced asymmetry in the *Is*PadC$^{Reg2}$ structure is characterized by one protomer in the P$_r$-state while the second protomer features a P$_{fr}$-conformation with its biliverdin chromophore in the 15*E* configuration and the associated PHY-tongue element rearranged into the characteristic P$_{fr}$ α-helical conformation. Obviously in the refolded state, the PHY domains exhibit greater flexibility, leading to a less defined electron density for this region compared to the rest of the protein. The directly linked PHY-tongue elements that are stabilized by interactions with the GAF domains, however, feature a well-defined electron density leaving no ambiguity regarding their differences in secondary structure (*Figure 2—figure supplement 1*). The P$_r$-state protomer, with its characteristic β-hairpin conformation in the tongue element, superimposes very well with the corresponding protomer of the *Is*PadC dark state structure (RMSD of 0.26 Å over 347 Cα-atoms for the PSM alignment for the respective chains B, *Figure 3—figure supplement 1*). The superposition is only disturbed by some flexibility towards the end of the coiled-coil linker and the associated DGC. The P$_{fr}$ protomer deviates from the P$_r$ state of wild-type *Is*PadC as well as the P$_r$ protomer of *Is*PadC$^{Reg2}$ by refolding of the PHY-tongue region into an α-helical structure that leads to a bending at the helical spine region triggering a quasi-translational movement of the associated PHY – coiled-coil – DGC region (RMSD of 1.56 Å over 402 Cα-atoms for the PSM alignment of *Is*PadC$^{Reg2}$ on *Is*PadC of the respective chains A). The directly linked GGDEF protomer repositions relative to the second DGC protomer increasing the asymmetry within the GGDEF dimer. By calculating the free energy of assembly dissociation (ΔG$^{diss}$) by PISA analysis (*Krissinel and Henrick, 2007*), we observed that the energetic stability of the PAS-GAF-PHY interface (residues 8–500) in the *Is*PadC$^{Reg2}$ structure is only slightly affected compared to the one calculated for *Is*PadC (*Gourinchas et al., 2017*) (ΔG$^{diss}$ = 6.5 kcal mol$^{-1}$ and 8.8 kcal mol$^{-1}$, respectively). Moreover, the stability of the PAS-GAF-PHY-coiled-coil assembly in the *Is*PadC$^{Reg2}$ structure is in the range of the same assembly calculated for the *Is*PadC structure (*Gourinchas et al., 2017*) (ΔG$^{diss}$ = 20.0 kcal mol$^{-1}$, and ΔG$^{diss}$ = 22.6 kcal mol$^{-1}$, respectively), which suggests that the overall stability of the phytochrome sensor is only slightly affected by the structural rearrangement of the PHY-tongue and the coiled-coil linker. A closer look at the dimeric PHY interface shows that upon linker helix translation the Arg327 - Asp494 intermolecular salt bridges observed in the dark state structure of *Is*PadC are absent in the *Is*PadC$^{Reg2}$ structure thereby enhancing the conformational freedom at the PHY interface (*Figure 3—figure supplement 1c*). Interestingly, the asymmetry of the *Is*PadC$^{Reg2}$ structure is also reflected in several other amino acids at the dimeric interface, for instance Phe132, Gln137, Arg140, Met320 for the PSM interface, and His577, K569, Y670 at the DGC interface. Upon isomerization of the biliverdin cofactor, Tyr168 and Phe195 change their rotamer conformation to accommodate the D-ring rotation as already described for other phytochrome species (*Yang et al., 2009*; *Anders et al., 2013*; *Burgie and Vierstra, 2014*). Also His193 changes its rotamer to interact with the carbonyl oxygen of the rotated D-ring (*Figure 3b–c*). In addition, the tryptophan switch scenario (*Anders et al., 2013*; *Burgie and Vierstra, 2014*) is also observed for refolding of the *Is*PadC PHY-tongue.

## DGC activity is regulated by structural rearrangements and changes in conformational dynamics

As far as downstream signaling of the activated PSM heterodimer is concerned, the crystal structure supports the switching mechanism between two coiled-coil registers proposed recently (*Gourinchas et al., 2017*). In contrast to the anticipated rotation mechanism, we observe a quasi-translational rearrangement (*Figure 3a,d*, and *Figure 3—figure supplement 1*) that is linked to a hinge movement of the whole PHY-coiled-coil linker region (*Figure 2d*). This type of functional rearrangement of the phytochrome dimer interface has also been observed by comparison of various phytochrome structures already at the PSM level (*Gourinchas et al., 2017*; *Yang et al., 2015*). More precisely, the refolding of the PHY-tongue is linked to a tilting of the central helical spine – PHY – coiled-coil region. This eventually causes the C-terminal parts of the coiled-coil linker to interact more tightly and repositions the directly following DXLT motifs at the dimer interface of the GGDEF domains. This highly conserved motif is directly linked to structural elements that are part of the open active site at the dimer interface of the GGDEF domain (*Schirmer, 2016*). In the absence of substrate, the coiled-coil linker rearrangement induces a different DGC dimer architecture due to a displacement of the DGC protomer linked to the photoactivated phytochrome protomer (*Figure 2c*). This results in a more open interface at one side of the dimer that potentially facilitates substrate binding or product release. In addition, we observed that the GGEEF motifs of *Is*PadC$^{Reg2}$ are positioned closer to each other, indicating that the two GTP substrates could be more closely interacting and potentially initiating the chemical reaction between the two molecules in an asymmetric fashion, as opposed to the symmetric binding of GTP in the dark inhibited conformation (*Gourinchas et al., 2017*). It should be noted, however, that the complex reaction mechanism of DGCs involves coordination of two GTP molecules, initial bond formation to form the asymmetric pppGpG intermediate, structural rearrangements to form the second linkage of c-di-GMP and subsequent product dissociation. Therefore, a single crystallographic snapshot cannot correspond to 'the active' conformation of DGCs; the dynamic nature of GGDEF dimerization (*Gourinchas et al., 2017*; *Chan et al., 2004*; *Schirmer, 2016*; *Zähringer et al., 2013*) rather supports a model in which the coiled-coil register switching influences the conformational sampling of different dimeric interfaces. In the case of the stimulating register 2 conformation of the coiled-coil, the sampling of dimers compatible with the geometry required for the DGC reaction to proceed is facilitated. Since an additional increase in activity can be achieved by stimulating the conformational dynamics of the coiled-coil by illumination, not a static dimer interface, but rather a dynamic assembly of GGDEF protomers, enables optimal DGC activity.

Interestingly, the superposition of the *Is*PadC$^{Reg2}$ structure with the GTP soaked *Is*PadC structure (*Gourinchas et al., 2017*) suggests that GTP binding within the DGC dimer induces a slight translation of the coiled-coil linker towards the stimulating register (*Figure 3—figure supplement 1a,b*). Therefore, the full-length system involves a complex cross-talk between sensor and effector domains that influence each other based on finely tuned dimer interfaces. The coiled-coil as central communication hub integrates the functional state of input and output modules, indicating that the rational design of artificial sensor-effector couples needs to consider this complex interplay of dynamic dimeric interfaces of sensor and effector domains as well as properties of the coiled-coil linker.

The importance of conformational dynamics was also confirmed by hydrogen-deuterium exchange coupled to mass spectrometry (HDX-MS) experiments (*Figure 4*, and *Figure 4—figure supplement 1*). Interestingly, phytochrome activation by red light results in similar effects for both *Is*PadC and *Is*PadC$^{Reg2}$. Both constructs showed an overall increase in deuterium uptake upon red light illumination indicating a light-induced increase of conformational dynamics. However, the relative amount of deuterium exchanged under dark and light conditions is lower for peptides in functionally important regions of *Is*PadC$^{Reg2}$ compared to *Is*PadC. This general decrease of conformational dynamics of *Is*PadC$^{Reg2}$ highlights the molecular cross-talk between different functionally important elements (*Gourinchas et al., 2017*). The decrease of deuterium uptake of coiled-coil peptides correlates with the stabilization of the linker in the stimulating register conformation as anticipated for the amino acid substitutions in *Is*PadC$^{Reg2}$. The fact that this variant has a higher turnover rate constant in the dark compared to *Is*PadC indicates that the coiled-coil conformation in the stimulating register has a positive effect on enzymatic activity of the DGC (*Figure 5a*). However, HDX-MS shows that illumination increases the dynamics of the coiled-coil and thereby destabilizes

the coiled-coil arrangement, which results in an additional increase in enzymatic activity; this implies that, in addition to the register conformation, the overall dynamics of the system play an important role in modulating DGC activity (*Figure 5a*).

Importantly, HDX-MS measurements derive from an ensemble measurement of all peptides in the protein dimer. Therefore, it is not possible to specifically address asymmetric stabilization or destabilization in the context of the full-length protein dimer in solution. However, the broad deuterium uptake distributions of PHY-tongue peptides upon illumination (*Figure 4e*) are indicative of a deuterium exchange regime that considers two conformational states with pronounced differences in deuterium uptake kinetics. Even if these observations are no direct proof of a heterodimer in solution, they suggest that different conformational states of the PHY-tongue elements co-exist upon light activation of the PadC system. This conformational coupling is also reflected in the crystal structure of *Is*PadC$^{Reg2}$ and its pronounced differences to the inactive wild-type structure. Even though the $P_{fr}/P_r$ heterodimer conformation is not the energetically most favorable in solution, it does represent one functional form of the dimeric assembly that highlights the cross-talk between the site of light activation and the central regulatory element for enzymatic activity – the coiled-coil linker. Additional experiments to address the directionality of signal transduction by producing apoproteins of *Is*PadC$^{Reg2}$ and wild-type *Is*PadC could provide interesting additional insights, however, the isolation of apoprotein is not feasible in the *Is*PadC system.

Figure 5 summarizes the observed influence of the coiled-coil register conformation and the dynamics of the system for regulating output activity of the DGC in a simplified model. To that end, the structure of *Is*PadC$^{Reg2}$ and the structural changes associated with activation will help to read-dress the activation mechanism of other phytochromes that are commonly used for designing optogenetic tools (*Tischer and Weiner, 2014*; *Shcherbakova et al., 2015*). In the context of phytochrome function, our study confirms that functional $P_{fr}$/meta-R or $P_{fr}/P_r$ phytochromes exist and that, while chromophore isomerization coupled to reorganization of the PHY-tongue region appears to be conserved in phytochromes, the extent to which this affects the output domain is tuned by the dimeric phytochrome interface via modulation of the effective population of $P_{fr}$-states as characterized by the structural rearrangements of the PHY-tongue and the PHY dimer interface. Ultimately, the flexibility to stabilize different functional states via the dimer interface enables phytochromes to be adapted to various effectors and also enables tuning of the action spectra by reducing the cross-section with inactivating far-red light.

## Materials and methods

### Protein preparation

Linker register variants were produced by site-directed mutagenesis following the protocol described by Liu *et al.* (*Liu and Naismith, 2008*) using pETM-11 *Is*PadC based plasmids as templates (*Gourinchas et al., 2017*) (used primers are listed in Table 3 in *Supplementary file 1*).

*Is*PadC variants were expressed and purified as described in detail previously (*Gourinchas et al., 2017*). Briefly, (His)$_6$-tagged holoproteins were expressed in BL21 (DE3) containing the previously generated pT7-ho1 helper plasmid (*Gourinchas et al., 2017*) to co-express heme oxygenase (HO-1). Cells grown at 37°C in LB medium completed with 0.36% (w/v) glucose and 10 mM MgCl$_2$ were cooled to 16°C and the medium was supplemented with 10 mg liter$^{-1}$ δ-aminolevulinic acid prior to induction with 0.25 mM isopropyl-β-D-thiogalactopyranoside (IPTG) for ~15 hr. After cell harvesting, the soluble lysates obtained by combined treatment with lysozyme (100 μg ml$^{-1}$) and sonication (2 × 5 min, 100 W, continuous mode; Labsonic L, Sartorius, Göttingen, Germany) were separated by ultracentrifugation (206,000 g), and affinity-purified on a Ni$^{2+}$-Sepharose matrix (Ni Sepharose 6 Fast Flow, GE Healthcare, Little Chalfont, UK). After a washing step (lysis buffer containing 30 mM imidazole), proteins were eluted (lysis buffer containing 250 mM imidazole) and dialyzed overnight at 4°C in presence of histidine-tagged tobacco etch virus (TEV) protease (~1:16 TEV/substrate). Afterwards TEV and cleaved histidine tag were removed from the protein solution by reloading the dialyzed protein onto the Ni$^{2+}$ column and collecting the flow through. An additional step of size-exclusion chromatography on a 16/60 Superdex 200 prep grade column (GE Healthcare) was performed before proteins were concentrated in the buffer listed in Table 3 in in *Supplementary file 1*

(Amicon Ultra-15; Merck Millipore, Burlington, MA), aliquoted, flash-frozen in liquid nitrogen and conserved at −80°C until needed.

For characterization of steady-state kinetics of the IsPadC variants, the proteins were subject to an additional size-exclusion chromatography run with a 10/300 Superdex 200 increase analytical grade column (GE Healthcare) in order to exchange the protein into HPLC buffer (Table 3 in in *Supplementary file 1*) and to remove potential protein aggregates caused by sample thawing.

## Cell-based diguanylyl cyclase assay

To screen for diguanylyl cyclase activity, we followed our previously described protocol (*Gourinchas et al., 2017*) adapted from the method described by Antoniani et al. (*Antoniani et al., 2010*). *E. coli* BL21 (DE3) co-expressing the pT7-ho1 plasmid and transformed with pETM-11 IsPadC based plasmids were cultivated at 30°C to $OD^{600}$ 0.5 in YESCA medium (casamino acids 10 mg ml$^{-1}$ and yeast extract 1.5 mg ml$^{-1}$) supplemented with $MgSO_4$ (0.05 mg ml$^{-1}$), $FeSO_4$ (0.5 µg ml$^{-1}$), kanamycin (30 µg ml$^{-1}$), and chloramphenicol (34 µg ml$^{-1}$). After 4 hr induction at 16°C with 0.25 mM IPTG in the presence of 10 mg liter$^{-1}$ δ-aminolevulinic acid, 3 µl of the induced culture were spotted on YESCA agar plates containing 30 µg mL$^{-1}$ kanamycin, 34 µg mL$^{-1}$ chloramphenicol and 0.01 mg mL$^{-1}$ congo-red, and incubated at 20°C for 16 hr in the dark or under constant red light illumination (630 nm, 75 µW cm$^{-2}$). As negative control we used a pETM-11 AppA construct (*Winkler et al., 2013*) that does not show any DGC activity.

## Spectroscopic characterization

UV-Vis absorption spectra were acquired with a CCD-based Specord S300 Vis spectrophotometer (Analytic Jena, Jena, Germany) using protein samples equilibrated at 20°C and diluted in their corresponding storage buffers (Table 3 in *Supplementary file 1*). Dark-adapted $P_r$-state absorption spectra were measured under non-actinic conditions by minimizing the contact time with measuring light using a neutral density filter (ND = 1.0; Thorlabs, Newton, NJ) between the light source and the sample cuvette. $P_{fr}$-enriched spectra were recorded under constant red light irradiation (660 nm, 45 mW cm$^{-2}$, Thorlabs) in the presence of the same neutral density filter for the measuring light.

To record changes in $P_{fr}/P_r$ ratio under various photon flux densities, the red LED (660 nm, Thorlabs) was adjusted to different light intensities at a distance of 60 cm from the sample. After establishing a steady-state at each light intensity, spectra were measured and absorbance values at 750 nm were extracted. Each light intensity measurement was acquired from the corresponding lower to the maximal intensity setting before allowing the sample to recover to its ground state and repeating the measurement in a technical triplicate. The sample standard deviation of individual measurements is used as error indicator in *Figure 1d*.

$P_r$-state recovery kinetics were followed at 710 and 750 nm after 1 min red light illumination (660 nm, 45 mW cm$^{-2}$, Thorlabs) using a Specord 200 Plus spectrophotometer (Analytic Jena) with 10 ms integration time sampled every 5 s and the recorded time constants differ by less than 10% between 700 and 750 nm measurements.

To record the UV-Vis absorption spectra of the denatured biliverdin bound proteins, we followed the previously described method from Thümmler et al. (*Thümmler et al., 1981*). The concentrated protein sample was kept in the dark, or irradiated 1 min with saturating red light (660 nm, 45 mW cm$^{-2}$, Thorlabs), before diluting it 1:10 in quenching buffer (0.1% trichloroacetic acid in methanol) to a final concentration of 1–2 µM. The absorption spectra of the denatured $P_r$- or $P_{fr}$-states were then directly recorded. Re-illumination of $P_{fr}$-peptides by red light triggers the isomerization of biliverdin back to the $P_r$-state and therefore $P_{fr}$-peptides were denatured directly after switching off the red light source and kept in the dark until their measurement.

Fluorescence measurements were performed using an RF-5301 PC spectrofluorophotometer (Shimadzu, Kyoto, Japan) equipped with a concave blazed holographic grating excitation and emission monochromator, and containing a 150 W Xenon lamp (Ushio, Cypress, CA) as light source. Samples diluted to 2 µM were equilibrated at room temperature and complete darkness for the dark state measurement or 1 min under saturating red light (660 nm, 45 mW cm$^{-2}$, Thorlabs) for the light state measurement prior to data acquisition. A sampling interval of 0.2 nm was used with a slit opening of 1.5 nm for the excitation slit and 5 nm for the emission slit.

## Crystallization and structure elucidation

The $Is$PadC$^{Reg2}$ variant was crystallized at 293 K under dark conditions using a sitting-drop vapor diffusion setup. To obtain the best diffracting crystals, 2 µl drops containing equal volumes of protein solution at 5 mg ml$^{-1}$ and reservoir solution (0.1 M magnesium formate, 12% (w/v) PEG 3,350), were seeded using the method of streak seeding and equilibrated against 35 µl of reservoir solution. Crystals appeared after overnight incubation and reached final dimensions within 10 days.

Crystals were harvested under low intensity non-actinic green light conditions (520 ± 20 nm LED) by transferring the crystals to a cryprotectant solution (reservoir solution containing 30% glycerol) and subsequent flash-freezing in liquid nitrogen. Diffraction data for $Is$PadC$^{Reg2}$ were collected at beamline P11 of the Deutsches Elektronen-Synchrotron (DESY). Data were processed using the XDS program package (*Kabsch, 2010*) (Table 2 in *Supplementary file 1*).

The crystal structure of $Is$PadC$^{Reg2}$ variant was solved by molecular replacement using PHENIX Phaser (*McCoy et al., 2007*) with the PAS-GAF fragment and the PHY-domain lacking the tongue extensions of the full-length structure of $Is$PadC (pdb 5LLW), as well as the DGC fragment of the $Is$PadC GTP soaked structure (pdb 5LLX). Two molecules for the PAS-GAF, PHY, and DGC search model were successfully placed in the asymmetric unit. The initial model of a PAS-GAF-PHY-GGDEF dimer was then manually extended to build the PHY-tongue and coiled-coil linker regions in several rounds of maximum-likelihood refinement of models modified with Coot using $\sigma_A$-weighted $2mF_o - DF_c$ and $F_o - F_c$ electron density maps. In addition, torsion-NCS restraints and secondary structure restraints were included together with TLS groups for the individual domains of the protein. During the final rounds of refinement reference model restraints were applied on PAS-GAF-PHY excluding the PHY β-hairpin extension using a previously solved high resolution $Is$PadC PSM structure (*Gourinchas et al., 2017*), and on GGDEF domains using the $Is$PadC dark state structure (*Gourinchas et al., 2017*) and optimization of X-ray and ADP weights was performed.

Spectral properties of the crystal were recorded at 100 K under a cryostream using a microspectrophotometer at the cryobench ID29S of the ESRF facilities (*von Stetten et al., 2015*).

## Kinetic analysis

To record the conversion of GTP to c-di-GMP for all the variants, we adapted a protocol (*Gourinchas et al., 2017*) for high performance liquid chromatography (HPLC) from a previously described method (*Enomoto et al., 2014*). Briefly, purified protein samples were mixed with GTP at various concentrations in reaction buffer (50 mM Hepes pH7.0, 500 mM NaCl, and 50 mM MgCl$_2$) at 20°C under non-actinic light for the dark state measurement, or under constant red light illumination (630 nm, 0.7 mW cm$^{-2}$, Luminea LED) following a 1 min pre-illumination of the sample for the light state measurements. Samples were then thermally inactivated by 1 min incubation at 95°C. After separating the substrate and products from denatured protein by centrifugation, the nucleotides were separated by a linear 7 min gradient from 2% to 20% MeOH using a reversed phase HPLC column (SunFire C18 4.6 × 100; Waters, Milford, MA) equilibrated in 10 mM triethylammonium formate (pH 6.0). For protein variants isolated in a partially activated state and featuring a slow thermal recovery a prolonged incubation of the sample under far-red light (730 ±20 nm, Thorlabs LED) coupled to a far-red light bandpass filter (750 nm, Thorlabs) at room temperature and complete darkness was performed to fully populate the P$_r$-state of the sample prior to the dark state measurements. All kinetic data were normalized to the concentration of the dimeric protein, and all samples were corrected for the amount of c-di-GMP formed during the inactivation step.

## Hydrogen-Deuterium exchange coupled to mass spectrometry

Deuterium labeling experiments were performed to address the effect of illumination on the conformational dynamics of $Is$PadC$^{Reg2}$ in order to compare it with the previously performed analysis of the $Is$PadC system (*Gourinchas et al., 2017*). Briefly, 200 µM protein sample were equilibrated at 20°C in the dark or under red light illumination (630 nm, 0.7 mW cm$^{-2}$, Luminea LED) for 1 min prior to starting the deuterium-labelling reaction by diluting the sample 20-fold in D$_2$O containing 10 mM Hepes (pD 7.0), 150 mM NaCl, and 10 mM MgCl$_2$. Reactions were quenched with ice-cold 200 mM ammonium formic acid (pH 2.5) after 10 s, 45 s, 3 min, 15 min, and 60 min and injected into a cooled HPLC setup as described previously (*Winkler et al., 2015*). To sum up, deuterated samples were digested on an immobilized pepsin column (Poroszyme; Applied Biosystems, Foster City, CA)

operated at 10°C. Resulting peptides were desalted on a 2 cm C18 guard column (Discovery Bio C18, Sigma), and separated during a 7 min acetonitrile gradient (15–50%) in the presence of 0.6% (v/v) formic acid on a reversed phase column (XR ODS 75 x 3 mm, 2.2 µM, Shimadzu). Peptides were then infused into a maXis electrospray ionization-ultra high resolution-time-of-flight mass spectrometer (Bruker, Billerica, MA). Deuterium incorporation was analyzed and quantified using the Hexicon 2 software package (http://hx2.mpimf-heidelberg.mpg.de) (*Lindner et al., 2014*).

## Acknowledgements

We thank E Zenzmaier for technical support and U Vide for cloning, expressing and purifying the *Is*PadC$^{\Delta442-477::SG}$ variant. We are grateful to I Schlichting for continuous support. We thank R Lindner for input during the HDX-MS measurements and during data analysis with the Hexicon two software package. We are grateful to K Gruber for providing access to the crystallization facilities at the Institute of Molecular Biosciences at the University of Graz. We appreciate the efforts of the Strubi-Graz beamline teams that are supported by BAG proposals MX-526 and MX-1939 at various beamlines. Parts of this research were carried out at PETRA III at DESY (*Burkhardt et al., 2016*), a member of the Helmholtz Association (HGF). We would like to thank A Burkhardt for assistance in using beamline P11. We are also grateful to the support of the ESRF staff at the cryobench facility ID29-S (*von Stetten et al., 2015*), specifically A Royant and G Gotthard. The authors also acknowledge the platforms of the Grenoble Instruct center (ISBG; UMS 3518 CNRS-CEA-UJF- EMBL) supported by the French Infrastructure for Integrated Structural Biology Initiative FRISBI (ANR-10-INSB-05–02) and GRAL (ANR-10-LABX-49–01) within the Grenoble Partnership for Structural Biology (PSB). Atomic coordinates and structure factors for the reported crystal structure have been deposited with the Protein Data Bank under accession number 6ET7. GG. is supported by the by the Austrian Science Fund through the PhD programme 'DK Molecular Enzymology' (W901). The authors gratefully acknowledge support from NAWI Graz. AW. acknowledges funding by the Austrian Science Fund (FWF): P27124.

## Additional information

### Funding

| Funder | Grant reference number | Author |
| --- | --- | --- |
| Austrian Science Fund | P27124 | Andreas Winkler |

The funders had no role in study design, data collection and interpretation, or the decision to submit the work for publication.

### Author contributions

Geoffrey Gourinchas, Data curation, Formal analysis, Investigation, Visualization, Methodology, Writing—original draft, Writing—review and editing; Udo Heintz, Resources, Investigation, Methodology, Writing—review and editing; Andreas Winkler, Conceptualization, Resources, Formal analysis, Supervision, Funding acquisition, Validation, Investigation, Methodology, Project administration, Writing—review and editing

### Author ORCIDs

Geoffrey Gourinchas  http://orcid.org/0000-0003-2543-3518
Andreas Winkler  http://orcid.org/0000-0001-6221-9671

### Decision letter and Author response
Decision letter https://doi.org/10.7554/eLife.34815.025
Author response https://doi.org/10.7554/eLife.34815.026

## Additional files

### Supplementary files
• Supplementary file 1.
DOI: https://doi.org/10.7554/eLife.34815.019
• Transparent reporting form
DOI: https://doi.org/10.7554/eLife.34815.020

### Data availability
Diffraction data have been deposited in the PDB under the accession code 6ET7.

The following dataset was generated:

| Author(s) | Year | Dataset title | Dataset URL | Database, license, and accessibility information |
|---|---|---|---|---|
| Gourinchas G, Winkler A | 2017 | Activated heterodimer of the bacteriophytochrome regulated diguanylyl cyclase variant - S505V A526V - from Idiomarina species A28L | http://www.rcsb.org/pdb/search/structid-Search.do?structureId=6ET7 | Publicly available at the RCSB Protein Data Bank (accession no: PDB 6ET7) |

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
