## [Decision Letter]

Thank you for submitting your article "Asymmetric activation mechanism of a homodimeric red light regulated photoreceptor" for consideration by *eLife*. Your article has been reviewed by three peer reviewers, and the evaluation has been overseen by a Reviewing Editor and Michael Marletta as the Senior Editor. The following individuals involved in review of your submission have agreed to reveal their identity: Katrina T. Forest (Reviewer #2).

The reviewers have discussed the reviews with one another and the Reviewing Editor has drafted this decision to help you prepare a revised submission.

Summary:

The manuscript by Gourinchas, Heintz and Winkler addresses the mechanistic role of structural asymmetry in allosteric activation of modular signaling proteins. Specifically, the authors examined the light signaling mechanism in a full-length photo-activated diguanylyl cyclase, *Is*PadC. They reported the crystal structure of an activated *Is*PadC variant (*Is*PadC^Reg2^). By comparing the IsPadCreg2 structure with the inactive wild type *Is*PadC structure from their previous work, they propose that light-induced structural asymmetry represents a critical event in *Is*PadC activation. Apparently, the output module can be de-coupled from the photosensing module and remain either inactive or active, and one half of the molecule can be structurally P_r_, while the other half is structurally P_fr_.

Relatively local changes in the dimer interface are seen relative to the native structure, supporting the conclusion that there are not any major conformational changes upon activation but rather subtle rearrangements at interfaces. HDX experiments are also presented and provide a complete comparison to the native protein results presented previously. These results support the model presented and confirm the overall structure of Reg2 behaves very similarly to the native protein in terms of accessibility changes before and after illumination.

While the topic and findings of this work are interesting, several major issues must be addressed or clarified before its final publication. All three reviewers were in agreement about the major strengths and weaknesses of the paper, and this decision letter uses wording from each of the reviews and the discussion in an attempt to summarize the reviewers' comments.

Essential revisions:

1) The most significant concern of all reviewers and the major point of our discussion is the inconclusive interpretation of the spectroscopic data regarding structural asymmetry at the chromophore site. The spectroscopic arguments lack validity because there are too many equally valid alternatives, including the possibility of apo protein and/or misfolded protein within the sample. It is not clear if the authors mean the P_fr_/P_r_heterodimer or P_fr_/meta-R heterodimer represents the functional dimer for *Is*PadC. It is important to distinguish between the signaling state of the protein structure and the signaling state of the chromophore, because the protein tertiary structure may or may not correspond to a specific chromophore conformation in photoreceptors. Very often, they are decoupled in asymmetric dimers such as *Is*PadC or in crystals due to lattice constraints. The authors should both provide more detailed information (see below) and tone down interpretations about 50%/50% Z/E conformation in the activated Reg2 protein.

The following should be included: a) P_r_-P_fr_ difference spectra for Reg2 in solution.

b) More careful presentation of P_fr_ to P_r_ thermal reversion data.

It appears from the Materials and methods section that this reaction was measured at 700 and 750 nm, but Supplementary file 1 indicates in the legend and the table that it was measured at 710 nm. Thermal reversion could be used in a more systematic way to address the possibility of a non-P_fr_photoproduct. The authors noted that three exponentials are required to explain P_r_ accumulation. Are there three exponentials for P_fr_ depletion? If so, do the time constants of these exponentials match? And, if so, are the ratios of the amplitudes at 750 nm to those at 700 nm similar for each time constant? If this is the case, the authors may just be looking at three manifestations of P_fr_ to P_r_ thermal reversion (e.g., P_fr_/P_fr_ to P_fr_/P_r_, P_fr_/P_r_ to P_r_/P_r_, thermal reversion from misfolded species, etc.). If not, the authors may have a non-P_fr_photoproduct (e.g., if only two exponentials are required for fitting of P_fr_ depletion, or if the 750 nm:700 nm amplitude ratio corresponding to one time constant differs greatly from the other two).

c) Simulated annealing omit-maps for LBV in subunit A and B as well as superposition of the bilin chromophores between subunit A and B in both the WT (inactive; symmetric) and Reg2 (activated; asymmetric) structure.

d) Superposition of the LBV chromophores from subunits A and B based on the structural alignment of the protein framework.

e) Simulated annealing omit-maps of the PHY-tongue in subunit A and B.

The crystal spectra are confusing rather than informative (Figure 1C). The authors appear to normalize spectra at the P_r_ state of the Q band of the crystalline and RL solution samples. However, proteins are structurally ordered within crystals, and random in a solution. Absorption spectra for a crystal will then be a function of the angle through which light passes. This issue is seen most clearly for the Soret peak, which is actually higher in magnitude than the Q band, this issue could also affect the relative extinction coefficients of the Q band of P_r_ and P_fr_, because the D-ring flips and the bilin moves within the binding pocket. The way to get around this would be to collect 180 degrees of absorption data for a single crystal and take a weighted average (weighted against the pathlength through the crystal), or more easily, collect spectra from several randomly oriented crystals in a cuvette.

2) Asymmetry in protein conformation:

Although there is an advantage in the same crystal form for WT and Reg2 (it implies this asymmetry can't be forced by the crystal packing), it is also possible that further conformational changes are inhibited by the crystal packing. If this asymmetric state is active, then the crystalline protein should be active. Whereas, if further protein changes are needed to get to a P_fr_/P_fr_ state for full activity, and if that is prohibited by crystal packing, the crystalline protein will be inactive. Unfortunately testing activity of crystals is tricky due to the possibility of soluble protein in equilibrium with crystalline. Thus, this assay is not a requirement of a revision, but it might be worth trying.

One suggestion for improved presentation of this point is that the authors should use superposition between subunit A and B (aligned according to PAS-GAF-PHY), rather than superposition between the WT and Reg2 structures (as shown in Figure 2 and Figure 3—figure supplement 3), to demonstrate structural asymmetry.

3) Translational Movement:

The conformational change that moves the linker coiled-coil into its second active register is described throughout the paper as a translation. This is not entirely accurate, as a true translation would mean that every atom has moved by the same x, y, z vector. Could "pivot" or "hinge" be used in the text, as it is in Figure 2D? This would more accurately describe motions which lead to larger translations of the atoms further from the pivot point than those close to it. (Examples in subsection “Detailed comparison of the active and inactive *Is*PadC crystal structures” and subsection “DGC activity is regulated by structural rearrangements and changes in conformational dynamics”).

4) Active site rearrangements:

It is a bit surprising and unacceptable that the authors did not compare any (asymmetric) structure features at the active site of DGC given the full-length WT and Reg2 structures. Ultimately, asymmetry in the tertiary/quaternary structure must serve the active site of DGC, in which the catalytic residues from each subunit are positioned to perform different (thus asymmetric) functional roles. In particular, direct comparison between the full-length crystal structures of WT and Reg2 is expected to reveal structural (re-)arrangements at the catalytic site of DGC. In the absence of structural data and discussion on asymmetry at the active site of DGC, the significance and contribution of the full-length structure is significantly diminished.

It is entirely possible that the catalytic site configuration in the crystal structure does not correspond to the signaling state of PSM. In other words, the effector domain is decoupled from the PAS-GAF-PHY domains, either due to the crystal packing effect or mutations in the linker helices. Regardless of the configuration of the DGC active site (presumably located at the dimer interface), the authors should take advantage of their full-length structures of WT and Reg2 with detailed descriptions and structural comparisons between the WT and Reg2 structures in the effector domains, in particular the active site.

5) Overall organization:

The entire Introduction is written as a single paragraph and is consequently pretty clunky. Also, the fusion of Results section and Discussion section causes confusion because the experimental data, structural interpretation, hypotheses, speculation and reasoning are not clearly distinguished in the text. Results section and Discussion section should be separated.

---

## [Author Response]

Essential revisions:1) The most significant concern of all reviewers and the major point of our discussion is the inconclusive interpretation of the spectroscopic data regarding structural asymmetry at the chromophore site. The spectroscopic arguments lack validity because there are too many equally valid alternatives, including the possibility of apo protein and/or misfolded protein within the sample.

After repeating some experiments and including additional controls, we are now in a position where we do understand in more detail what appears to be happening on a molecular level. Therefore, we agree that the initial spectroscopic characterization presented was superficial and to some extent not interpreted with the required care. We have now reanalyzed the data and provided additional figures/panels (Figure 1, Figure 1—figure supplement 2 and Figure 1—figure supplement 3) to enrich the presentation of the thermal recoveries in Supplementary file 1. Plotting the recoveries as A750/A710 ratios vs. time showed that extremely slow recovery contributions are present in *Is*PadC^Reg1^ and *Is*PadC^Reg2.a^. Including fixed, known endpoints (based on the initial dark state spectrum) for the fitting routine provided a much more robust observation of time constants and relative amplitudes of the various contributing species.

In fact, this revealed that all analyzed *Is*PadC constructs (starting from almost identical steady-state light-activated spectra) feature a fast recovering contribution with constant relative amplitude of roughly 36% at 750 nm. In combination with the observation that the recoveries proceed with a subtle non-isosbestic behavior (Figure 1—figure supplement 2E), this indicates a molecular species featuring a lower extinction coefficient at 750 nm with overlapping spectral properties to the P_r_ and P_fr_ states as being responsible for the fast recovery contribution. Further supporting our initial hypothesis of meta-R like states in combination with P_fr_ contributions being responsible for the steady-state light activated *Is*PadC spectrum, these spectral characteristics have been combined with references to previously published studies describing spectral properties of meta-R states (e.g. Borucki et al., 2005; subsection “Individual protomers of dimeric *Is*PadC feature different biliverdin environments upon illumination”). While we agree that other factors could also be responsible for the observed behavior, we think that the observations of asymmetry along several additional lines of evidence support our conclusion of an asymmetric heterodimer upon light activation. Especially the spectral characterization of the *Is*PadC^Reg2.a^ variant, with an extremely stable P_fr_ contribution, indicates that the stabilization of the coiled-coil in register 2 greatly enhances P_fr_ lifetime without affecting the amplitude of the fast recovering meta-R species. If the steady-state spectrum was obtained by mixed biliverdin populations in both cofactor pockets of the dimer or apo-protomer containing dimers, then such a pronounced stabilization of P_fr_ should result in an increased P_fr_ population in the steady-state, which is not the case.

The initial characterization of spectral properties during red light illumination clearly reveals a complex system involving multiple molecular species that are difficult to ultimately confirm with the methods used in this manuscript. More sophisticated experiments aiming at improving the time resolution and addressing vibrational modes of the biliverdin cofactor have been initiated in collaborative efforts and will help to address spectroscopic properties in more detail. Especially since no fast spectroscopy data are available for the *Is*PadC system, we refrained from a direct assignment of the meta-R state (meta-Ra or meta-Rc) but based on the structural implications we think that an early meta-R state where no tongue rearrangement has taken place is the most likely candidate to be stabilized in the asymmetric *Is*PadC heterodimer.

It is not clear if the authors mean the P_fr_/P_r_heterodimer or P_fr_/Meta-R heterodimer represents the functional dimer for IsPadC. It is important to distinguish between the signaling state of the protein structure and the signaling state of the chromophore, because the protein tertiary structure may or may not correspond to a specific chromophore conformation in photoreceptors. Very often, they are decoupled in asymmetric dimers such as IsPadC or in crystals due to lattice constraints.

We apologize for not being consistent enough with our description of conformational states describing the biliverdin environment, the PSM rearrangements, the coiled-coil register, or overall structural changes. Since this is complicated by the fact that some uncoupling of the structural rearrangements is always possible a clearer description of what is meant at which stage of the Results section and Discussion section is of course very important. We have tried to meet this requirement throughout the revised version, but have to admit that, considering the dynamic contribution of all functionally important elements, this can sometimes be difficult or even ambiguous. To reduce the ambiguity with respect to photoactivation, active register and enzyme activity, we have changed the nomenclature of the coiled-coil registers from inactive and active to inhibiting and stimulating, respectively.

With respect to the P_fr_/P_r_ vs. P_fr_/meta-R differentiation, we think that the global structural implications will be quite similar. Considering that early meta-R states are characterized by an isomerized biliverdin cofactor where only side chain rearrangements close to the biliverdin are realized without fully proceeding to the true P_fr_ state (i.e. including tongue refolding), we believe that the P_fr_/P_r_heterodimer crystal structure (obtained in the dark) globally reflects the structure of the P_fr_/meta-R heterodimers that are present during illumination with saturating red light. After switching off the light the meta-R component converts back to P_r_ relatively quickly and the resulting P_fr_/P_r_heterodimer would still be expected to be more active than a P_r_/P_r_homodimer in the wild-type. In terms of the nomenclature used by the reviewers both forms are therefore “functional dimers” of the *Is*PadC system. If the remark referred to “the active” dimer, we have to say that there is not only a single conformation that is active. Due to the partial uncoupling of different functional elements in the PSM, the coiled-coil and also the DGC the global dynamic interplay influences the enzymatic activity and, similar to many other photoreceptors, *Is*PadC is clearly not an on/off switch.

As far as lattice restraints are concerned, of course they can play a role for trapping non-functional states and could therefore bias our conclusions; however, in our case the crystal lattices are almost identical between the Pr/Pr homodimer in the wild-type structure and the P_fr_/P_r_heterodimer in *Is*PadC^Reg2^. We believe that the driving force for the observed structural rearrangements is due to the population of register 2 even in solution (as indicated by HDX-MS). This register conformation stabilizes the P_fr_ state as also shown by the thermal recoveries (Supplementary file 1) and even in solution there appears to be an equilibrium of P_fr_ and P_r_ states – eventually the heterodimeric species crystallizes in the case of *Is*PadC^Reg2^. However, the fact that the P_r_homodimer is still more stable in solution than a P_fr_/P_r_ mix, indicates that indeed there might be some uncoupling between the linker and the biliverdin environment; the coiled-coil can exist in register 2 at the same time as both protomers biliverdin environments are in Pr. We also cannot rule out some switching of the coiled-coil back to register 1 even in this variant, which most likely would also be partially uncoupled from the tongue and biliverdin environments. Clearly the signal transduction pathways in phytochromes do not follow the domino principle but more the violin model of allosteric regulation referred to also in our initial *Is*PadC publication (*Gourinchas et al., 2017*).

The authors should both provide more detailed information (see below) and tone down interpretations about 50%/50% Z/E conformation in the activated Reg2 protein.

As stated above, we have rephrased many passages where a direct reference to an activated species and a specific structural conformation is mentioned. Clearly there is no strict requirement for a 50/50 *E/Z* ratio for optimal activity and this is definitely not what we intended to say. Under high light conditions the *E* conformation will be strongly enriched and the resulting P_fr_/meta-R species (*E/E*) are highly active. However, under some circumstances an *E/Z*heterodimer will also exist and can be fully active considering the coupling of chromophore and coiled-coil environments. For example, under low light conditions – or even single photon excitation – the P_fr_/P_r_ species will dominate the equilibrium of species in solution. Also, after switching off the light the meta-R species recovers more quickly than P_fr_ and we again have a P_fr_/P_r_ enrichment. In both cases *E/Z* ratios of individual (activated) dimers will be close to 50% and dependent on the degree of coupling between the conformational states in functionally important elements. The population average might of course differ from that and due to the thermal recoveries also change in a time dependent manner.

If this comment was meant to refer only to the crystal structure, then I hope that the new data included (comparison of crystal spectra and OMIT maps – Figure 2—figure supplement 1, Figure 2—figure supplement 2 and Figure 2—figure supplement 3) further support the refined model of the *Is*PadC^Reg2^heterodimer with a 50% *E/Z* ratio.

To bring more clarity on these points, the following new data have been included as suggested by the reviewers:

The following should be included: a) P_r_-P_fr_ difference spectra for Reg2 in solution

In order to be consistent for all the variants described in the manuscript we included difference spectra of light–dark (which appears to be the more prevalent type of phytochrome difference spectra in the literature) for *Is*PadC, *Is*PadC^Reg1^, *Is*PadC^Reg2^, *Is*PadC^Reg2.a^ and the tongue deletion variant (*Is*PadC^D442-477::SG^). These spectra are now included in the new Figure 1—figure supplement 2 and Figure 1—figure supplement 3. Figure 1—figure supplement 2F shows a direct comparison of the individual difference spectra.

b) More careful presentation of P_fr_ to P_r_ thermal reversion data.It appears from the Materials and methods section that this reaction was measured at 700 and 750 nm, but Supplementary file 1 indicates in the legend and the table that it was measured at 710 nm. Thermal reversion could be used in a more systematic way to address the possibility of a non-P_fr_photoproduct. The authors noted that three exponentials are required to explain P_r_ accumulation. Are there three exponentials for P_fr_ depletion? If so, do the time constants of these exponentials match? And, if so, are the ratios of the amplitudes at 750 nm to those at 700 nm similar for each time constant? If this is the case, the authors may just be looking at three manifestations of P_fr_ to P_r_ thermal reversion (e.g., P_fr_/P_fr_ to P_fr_/P_r_, P_fr_/P_r_ to P_r_/P_r_, thermal reversion from misfolded species, etc.). If not, the authors may have a non-P_fr_photoproduct (e.g., if only two exponentials are required for fitting of P_fr_ depletion, or if the 750 nm:700 nm amplitude ratio corresponding to one time constant differs greatly from the other two).

Again, many thanks for bringing up this point and for the helpful suggestions. How our efforts to address these points have helped to improve our understanding of the system has already been stated in our reply to the general paragraph of comment 1. Therefore, we only refer to specific details at this point:

The thermal reversion data have been measured at the maximum of the P_r_ Q-band absorption of wild-type (710 nm) and the maximum of the P_fr_ (+ early meta-R) contribution in the light-dark difference spectrum of *Is*PadC (750 nm). We apologize for the typo in the Materials and methods section.

The time constants of individual exponential contributions match very well between 710 and 750 nm data. In addition to the time constants, also the amplitude contributions at the two wavelengths appear very similar. Detailed analysis of all phases resulted in amplitude@750/amplitude@710 ratios of: *Is*PadCt1 – 0.69, t2 – 0.62; *Is*PadC^Reg1^ t1 – 0.53, t2 – 0.75, t3 – 0.83; *Is*PadC^Reg2^ t1 – 0.64, t2 – 0.72; *Is*PadC^Reg2.a^ t1 – 0.66, t2 – 0.66, t3 – 0.7. Combining these data results in an average amplitude@750 over amplitude@710 ratio of 0.68, with a sample standard deviation of 0.08 for the 10 phases reported in Supplementary file 1. Along the lines of argumentation of the reviewers these observations would suggest different manifestations of P_fr_ to P_r_ recovery. In the case where two exponentials are sufficient to describe the data, the fact that the amplitude contributions are not 50%, however, suggests that the colored species of the steady-state light activated spectrum have different spectral contributions. Therefore, the most likely explanation in our opinion is the inclusion of an early meta-R species that has a lower extinction coefficient as P_fr_. The amplitude contributions of meta-R and P_fr_ species match very well with data provided for the corresponding states in Agp1 phytochrome (Borucki et al., 2005). Considering also the different isosbestic points of meta-R and P_fr_ with the P_r_ state, the observed non-isosbestic behavior (Figure 1—figure supplement 2E) matches the postulated behavior for a fast meta-R recovering species and a slower P_fr_ recovery. Looking more closely at the amplitude contributions in the range of the isosbestic points also revealed a strong dependence of relative amplitudes on the wavelength of the fit (*cf*. Author response image 1). Therefore, the similar amplitude contributions observed at 710 and 750 nm are merely a coincidence and do not indicate that our observations refer to different manifestations of P_fr_ recoveries.

But why do we need three exponentials for some constructs and only two to adequately fit the data for other constructs? This is a complex question and we do not have a conclusive answer to that; however, since the number of exponentials needed to describe the recoveries depends on salt and protein concentration, one potential answer would be that the more stable registers 1 and 2.a variants have different in solution properties than the other tested variants at the same protein concentration – e.g. tendency for unspecific oligomerization. Another explanation could be that the restriction of conformational dynamics in the coiled-coil linker affects the biliverdin environments of the two protomers differently. *Is*PadC^Reg2.a^ would favor the P_fr_ state on the structurally predestined protomer and only a small partially uncoupled subpopulation could form P_fr_ on the other protomer. Due to the restrained dynamics of the overall system the two different P_fr_–coiled-coil substructures would show different recoveries, which is why the total P_fr_ recovery contribution is split up into two different phases.

c) Simulated annealing omit-maps for LBV in subunit A and B as well as superposition of the bilin chromophores between subunit A and B in BOTH the WT (inactive; symmetric) and Reg2 (activated; asymmetric) structure

We performed OMIT map calculations using a recently described, improved version of OMIT map generation that handles issues with bulk-solvent models more adequately. Details can be found in Acta Crystallogr D Struct Biol. 2017 Feb 1;73(Pt 2):148-157 doi: 10.1107/S2059798316018210. While similar to simulated annealing OMIT maps calculated by a classical approach, the slightly improved quality of the “polder maps” enable easier visualization of features of interest. The LBV maps for both protomers in the asymmetric unit are shown in Figure 2—figure supplement 2 for both the *Is*PadC^Reg2^ structure (panel a and b) and wild-type *Is*PadC data (panel c and d). While the two biliverdin conformations of the wild-type structure are virtually identical, the two LBV models of the register 2 variant differ significantly. To illustrate the differences, the refined models of each protomer are overlaid with the model of the corresponding other protomer in the colors used in previous figures; i.e. dark-red for the biliverdin cofactor of the P_fr_protomer A and light-red for the P_r_protomer. We refrained from including the analogous superposition for the wild-type panels since it causes confusion due to the almost identical conformations in both protomers. However, panels e and f of Figure 2—figure supplement 2 show the biliverdin cofactors of both protomers based on the alignment of the PAS-GAF cores in the context of neighboring residues.

d) Superposition of the LBV chromophores from subunits A and B based on the structural alignment of the protein framework;

The superposition of the biliverdin cofactors mentioned above was calculated using the PAS-GAF bidomain for the structural alignment; we apologize if we have not correctly interpreted the lines in comments *c* and *d*, but we think that one superposition of the LBV cofactors is sufficient. To better appreciate the consequences of D-ring rotation in the near surrounding of the biliverdin, we have included panels e and f in Figure 2—figure supplement 2.

e) Simulated annealing omit-maps of the PHY-tongue in subunit A and B.

They have been included in Figure 2—figure supplement 1, using the same “polder maps” described in our response to comment *c*. While it is difficult to find suitable views that satisfy the reader in every aspect of both tongue conformations, we hope that the illustration chosen helps to appreciate the pronounced differences in tongue architectures. We believe that the interested reader will download the coordinates along with the deposited structure factors.

The crystal spectra are confusing rather than informative (Figure 1C). The authors appear to normalize spectra at the P_r_ state of the Q band of the crystalline and RL solution samples. However, proteins are structurally ordered within crystals, and random in a solution. Absorption spectra for a crystal will then be a function of the angle through which light passes. This issue is seen most clearly for the Soret peak, which is actually higher in magnitude than the Q band, this issue could also affect the relative extinction coefficients of the Q band of P_r_ and P_fr_, because the D-ring flips and the bilin moves within the binding pocket. The way to get around this would be to collect 180 degrees of absorption data for a single crystal and take a weighted average (weighted against the pathlength through the crystal), or more easily, collect spectra from several randomly oriented crystals in a cuvette.

The intention of this figure was to illustrate that there are spectral similarities between the *Is*PadC^Reg2^ crystals and the in solution properties upon illumination of all *Is*PadC constructs. The normalization to the Q band of in solution data, used to bring them on similar scales, was probably pushing the limits of acceptable data treatments too far; we apologize for that.

We now decided to strictly separate crystal spectra from in solution spectra and have therefore moved the crystal spectrum to Figure 2—figure supplement 3. To partially account for the extinction coefficient variation at different orientations of the crystal, we averaged 20 different snapshots at different orientations (knowing that they might be biased to suitable orientations of the plate like crystals in the loop relative to the incident light beam) acquired at cryogenic temperature. This is then compared to cryogenic data of P_r_/P_r_homodimer wild-type *Is*PadC crystals (4 different orientations available). These two datasets were normalized to the region where light and dark-state spectra intersect in solution (strictly speaking no isosbestic point is available, but for a direct comparison this is the best normalization approach that we could think of). In this illustration, it is obvious that *Is*PadC^Reg2^ crystals have a pronounced 750 nm contribution and that wild-type vs *Is*PadC^Reg2^ crystal spectra mimic the dark vs light spectra in solution, with the only exception that the Q band features are more defined at the cryogenic temperature measurements.

Of course, it was not our intention to say that the crystal spectrum is identical to the in solution spectrum of light activated *Is*PadC. Crystals clearly feature a mix of P_fr_/P_r_ species whereas in solution and at saturating red light conditions we expect P_fr_/meta-R heterodimers.

2) Asymmetry in protein conformation:Although there is an advantage in the same crystal form for WT and Reg2 (it implies this asymmetry can't be forced by the crystal packing), it is also possible that further conformational changes are inhibited by the crystal packing. If this asymmetric state is active, then the crystalline protein should be active. Whereas, if further protein changes are needed to get to a P_fr_/P_fr_ state for full activity, and if that is prohibited by crystal packing, the crystalline protein will be inactive. Unfortunately testing activity of crystals is tricky due to the possibility of soluble protein in equilibrium with crystalline. Thus, this assay is not a requirement of a revision, but it might be worth trying.

We have not performed any activity assays with isolated *Is*PadC or *Is*PadC^Reg2^ crystals. The reason for this relates to the caveats mentioned by the reviewers, but also the fact that the GGDEF mechanism is a lot more complex than substrate conversion at a defined active center. The DGC active site is formed at the interface of both subunits and requires concerted structural rearrangements of both subunits for appropriately positioning the two GTP molecules for initial bond formation (pppGpG intermediate), subsequent second bond formation (c-di-GMP), product and sideproduct(s) (PPi) dissociation, followed by binding of new GTP molecules. Considering the dynamic nature of the GGDEF interface and the reduced degrees of freedom in the crystal lattice, we would not expect that substantial turnover can happen in the crystals.

However, we tried soaking of *Is*PadC^Reg2^ crystals with GTP to see if we could obtain indications of initial steps of substrate conversion in the crystals. However, the crystals start cracking and partially disintegrate resulting in rather lousy diffraction data (worse than 4 Å). This could result in increased populations of soluble protein in equilibrium with the crystals and any activity data inferred from such measurements should not be correlated with the activity of the crystals. Interestingly, the same soaking procedure with the wild-type protein crystals did not result in cracking and a refined dataset has been included in our previous *Is*PadC publication (pdb 5LLX). This difference points at an increased structural flexibility of the *Is*PadC^Reg2^ system that might be correlated with structural arrangements that are not possible in the structure of the wild-type protein. In our opinion, the GTP induced structural changes, resulting in cracking of the crystals, are indicative of subsequent catalytic steps following GTP binding that are not feasible in the inhibiting register 1 conformation of the coiled-coil. Even though no full catalytic cycle is compatible with the lattice restraints, the fact that coiled-coil switching is coupled to P_fr_ formation on one protomer suggests that the observed P_fr_/P_r_ assembly is one functionally important state involved in DGC activity regulation. Whether the GGDEF conformations are fully coupled or partially uncoupled from the PSM-coiled-coil rearrangements in the crystal lattice is a different question. We tried to clarify these points in the Results section and Discussion section.

One suggestion for improved presentation of this point is that the authors should use superposition between subunit A and B (aligned according to PAS-GAF-PHY), rather than superposition between the WT and Reg2 structures (as shown in Figure 2 and Figure 3—figure supplement 3), to demonstrate structural asymmetry.

We fully agree that Figure 2 should focus only on the new *Is*PadC^Reg2^ data. Even though we mentioned that the PAS-GAF-PHY domains of the P_r_ conformation in the *Is*PadC^Reg2^ crystal structure are virtually identical to the wild-type P_r_ conformation, we have now replaced the corresponding grey cartoon representation in panel D of Figure 2. Due to the similarities of the P_r_ states, the new figure appears to only have a changed label compared to the original one. In terms of the superposition algorithm we refrained from using PAS-GAF-PHY, because in fact the PHY domain rearrangements relative to the PAS-GAF bidomain are one of the important things to show at this point (the PAS-GAF dimer remains virtually unaffected by P_fr_ state formation).

That the PHY dimer rearrangement is eventually also translated into an asymmetric GGDEF interface is reflected in panel c, where we now also only show *Is*PadC^Reg2^ data and aligned the GGDEF protomer of chain A to that of chain B. Since this no longer corresponds to the structural rearrangement induced by the coiled-coil transition, we have included an enriched version of the original comparison to the wild-type DGC interface in Figure 3—figure supplement 2. In the latter figure, we also kept the structural alignments between the wt / *Is*PadC^Reg2^ / *Is*PadC + GTP structures to highlight the differences in linker architectures induced by P_fr_ formation and GTP binding.

3) Translational Movement:The conformational change that moves the linker coiled-coil into its second active register is described throughout the paper as a translation. This is not entirely accurate, as a true translation would mean that every atom has moved by the same x, y, z vector. Could "pivot" or "hinge" be used in the text, as it is in Figure 2D? This would more accurately describe motions which lead to larger translations of the atoms further from the pivot point than those close to it. (Examples in subsection “Detailed comparison of the active and inactive IsPadC crystal structures” and subsection “DGC activity is regulated by structural rearrangements and changes in conformational dynamics”).

We admit that the term translation is only an approximation and only valid for the structural changes in the center of the coiled-coil region. The C-terminal PHY helix features a more pronounced hinge-like movement and also the C-terminal part of the coiled-coil follows a more complex structural reorganization to satisfy supercoiling of the two helices (strictly speaking, a pure hinge like movement in a coiled-coil would destroy the knobs into holes packing of this structural entity). In order to account for these complex structural rearrangements, we have now clearly stated in the Results section that the translational component in the coiled-coil is part of a hinge-like global structural rearrangement (subsection “The crystal structure of *Is*PadC^Reg2^ features an asymmetric Pfr/Pr heterodimer”). In fact, the hinge-like movement of the PHY-coiled-coil is coupled to the hinge movement of the PSM. The fact that the estimated pivot points do not overlap shows that even description as a hinge rearrangement is only an approximation in a complex three-dimensional system.

Therefore, we decided to use the term “quasi-translational” movement of the coiled-coil throughout the manuscript. This terminology allows clear differentiation from the more commonly assumed rotational coiled-coil transitions and still indicates that the situation is more complex than a pure translation. That “translation” is indeed a good approximation is also shown in a close up of the central coiled-coil linker part in Figure 3—figure supplement 1b.

4) Active site rearrangements:It is a bit surprising and unacceptable that the authors did not compare any (asymmetric) structure features at the active site of DGC given the full-length WT and Reg2 structures. Ultimately, asymmetry in the tertiary/quaternary structure must serve the active site of DGC, in which the catalytic residues from each subunit are positioned to perform different (thus asymmetric) functional roles. In particular, direct comparison between the full-length crystal structures of WT and Reg2 is expected to reveal structural (re-)arrangements at the catalytic site of DGC. In the absence of structural data and discussion on asymmetry at the active site of DGC, the significance and contribution of the full-length structure is significantly diminished.

We initially kept the discussion of mechanistic aspects of the DGC to a minimum because we wanted to focus the manuscript on the asymmetric activation mechanism of the phytochrome module. Our assumption was that including the complex DGC mechanism description in the manuscript would distract the reader from the major findings in the phytochrome part.

Especially since it is known that DGCs can crystallize in many (also unspecific) dimer interfaces, we are aware that any conclusions drawn from this specific crystallographic snapshot need to be considered with caution. Nevertheless, we have now included a basic discussion of the observed structural rearrangements and their potential implications for mechanistic aspects of GTP to c-di-GMP conversion (subsection “The crystal structure of *Is*PadC^Reg2^ features an asymmetric Pfr/Pr heterodimer”).

As far as asymmetric functional roles are concerned (as stated in the reviewers comment), we want to point out that asymmetry is likely important for the first bond formation step, but that in the second step of c-di-GMP formation a symmetric active site could also be envisaged (cf. pdb 3TVK with symmetry mate). Such multi-step pathways are difficult to rationalize with one crystallographic snapshot in the absence of substrate and therefore we tried not to overinterpret our findings in the corresponding parts of the Results section and Discussion section.

It is entirely possible that the catalytic site configuration in the crystal structure does not correspond to the signaling state of PSM. In other words, the effector domain is decoupled from the PAS-GAF-PHY domains, either due to the crystal packing effect or mutations in the linker helices. Regardless of the configuration of the DGC active site (presumably located at the dimer interface), the authors should take advantage of their full-length structures of WT and Reg2 with detailed descriptions and structural comparisons between the WT and Reg2 structures in the effector domains, in particular the active site.

As stated in our previous reply, we have included additional descriptions and comparisons between DGC conformations of WT and Reg2 in the revised version of the manuscript. We also clearly stated that the observed DGC arrangement has interesting implications for the catalytic mechanism but pointed out that the interpretation is complicated by the complex reaction mechanism and the fact that multiple rearrangements are expected to be involved in the turnover of GTP to c-di-GMP. This concept also shows that the dimeric DGC assembly cannot be strictly coupled to the coiled-coil conformation. Some uncoupling from the coiled-coil register conformation and ultimately the signaling state of the PSM is required to perform its function. An almost complete uncoupling from the PSM is nicely demonstrated by the *Is*PadC^Reg1^ and *Is*PadC^Reg2.a^ variants, which still show PSM photoactivation but no effect of light on the catalytic activity. In both variants the coiled-coil conformation is the activity determining parameter. The degree to which the PSM can influence the coiled-coil register and the degree to which this register is coupled to the DGC dynamics ultimately affect the dynamic range of the system.

5) Overall organization:The entire Introduction is written as a single paragraph and is consequently pretty clunky. Also, the fusion of Results section and Discussion section causes confusion because the experimental data, structural interpretation, hypotheses, speculation and reasoning are not clearly distinguished in the text. Results section and Discussion section should be separated.

We apologize for the lack of structure in our initial version and have introduced new paragraphs at appropriate positions throughout the manuscript. We also separated Results section and Discussion section, but kept the Discussion section very similar to the previous version. We removed redundant passages and pure results wherever possible but are aware that this was not possible for all aspects of the new Discussion section. We think that a certain degree of redundancy will help the reader to follow the main conclusions and believe that the overall readability has greatly improved from restructuring the initial manuscript.